# A Parallelized Database Damage Assessment Approach after Cyberattack for Healthcare Systems

Sanaa Kaddoura [1], Ramzi A. Haraty [2,*], Karam Al Kontar [2] and Omar Alfandi [1]

1    College of Technological Innovation, Zayed University, Abu Dhabi 144534, United Arab Emirates;
     sanaa.kaddoura@zu.ac.ae (S.K.); omar.alfandi@zu.ac.ae (O.A.)
2    Department of Computer Science and Mathematics, Lebanese American University,
     Beirut 1102 2801, Lebanon; karam.alkontar@lau.edu
*    Correspondence: rharaty@lau.edu.lb

**Abstract:** In the current Internet of things era, all companies shifted from paper-based data to the electronic format. Although this shift increased the efficiency of data processing, it has security drawbacks. Healthcare databases are a precious target for attackers because they facilitate identity theft and cybercrime. This paper presents an approach for database damage assessment for healthcare systems. Inspired by the current behavior of COVID-19 infections, our approach views the damage assessment problem the same way. The malicious transactions will be viewed as if they are COVID-19 viruses, taken from infection onward. The challenge of this research is to discover the infected transactions in a minimal time. The proposed parallel algorithm is based on the transaction dependency paradigm, with a time complexity $O((M+NQ+N^3)/L)$ (M = total number of transactions under scrutiny, N = number of malicious and affected transactions in the testing list, Q = time for dependency check, and L = number of threads used). The memory complexity of the algorithm is $O(N+KL)$ (N = number of malicious and affected transactions, K = number of transactions in one area handled by one thread, and L = number of threads). Since the damage assessment time is directly proportional to the denial-of-service time, the proposed algorithm provides a minimized execution time. Our algorithm is a novel approach that outperforms other existing algorithms in this domain in terms of both time and memory, working up to four times faster in terms of time and with 120,000 fewer bytes in terms of memory.

**Keywords:** damage assessment; information warfare; malicious transactions; transactions dependency

## 1. Introduction

Most of the applications of information systems, especially healthcare systems, are now based on online databases that contain huge amounts of data. The security of such databases is essential to ensure an information system that follows the CIA security model: confidentiality, integrity, and availability [1]. Information security involves all processes for protecting data and reducing the adverse effects of any incidents of unlawful use, disclosure, deletion, corruptions, or any form of misuse. As defined by [2], information security is the assurance that information risks and controls are in balance. Cryptographic hash functions are widely used in information security in many areas, like digital signatures and authentication, cybersecurity for risk management, and healthcare systems security [3–6].

Information warfare can be defined from different perspectives. In this paper, it is defined as an attack on a database to cause a denial of service of the information system [7], while defensive information warfare is the set of actions that protects the database from attacks.

The defensive information warfare paradigm consists of three main layers: prevention, detection, and recovery [8]. Preventive measures are the first line of defense. In this layer, techniques should be designed to guard hardware, software, and user data against threats

from both outsiders as well as malicious insiders. Although many security preventive measures are encountered in all healthcare sectors, statistics still prove that there is always a successful attack [8]. Hence, it is crucial to have detection measures to detect any tampering with data. Detection is usually done by the intrusion detection system (IDS). Once an IDS detects an attack, recovery should start to roll back any changes to ensure the integrity of the data. As the world moves toward the Internet of things (IoT) era, companies and organizations are replacing traditional paper-based data with new electronic data to achieve higher efficiency in processing. Online data provides availability and ease of sharing, but on the other hand, it poses privacy and security concerns.

This paper mainly focuses on healthcare systems, where data needs to be highly available to doctors and nurses to help them do their job, as downtime may be life-threatening to patients. In addition to the immediate availability of data, it should also be secure and correct. Any intentional or unintentional misinformation can jeopardize patients' health and, potentially, their lives. Information about patients, such as medication, allergies, surgeries, medical conditions, and medical history, is confidential and should only be available to authorized personnel. Hence, the CIA triad becomes essential to any implemented system. Last year, more than 8 billion malicious data attacks occurred [9]. In addition, 89% of healthcare organizations experienced data breaches in the past two years [9]. Healthcare databases are precious targets for attackers because they facilitate identity theft and cybercrimes. Any denial of service of healthcare databases affects patients' lives.

The most critical metric to be considered while securing healthcare databases is speed. A patient's history should be retrieved as soon as possible, and healthcare transactions should not be affected for a long time by an attack. Hence, database assessment for the healthcare system recovery should be efficient to prevent long downtimes while performing these procedures.

The objective of this paper is to develop an efficient algorithm to detect malicious transactions on a patient's data. Given the fact that a malicious transaction may not be immediately detected and hence may affect other transactions, dependency among transactions is taken into consideration by the proposed algorithm. The problem statement is further explained in Section 2. A parallel algorithm using multithreading is proposed, based on the transaction dependency paradigm. The time complexity of the serial version of the algorithm is $O(M+NQ+N^3)$, where M is the total number of transactions, N is the number of insertions in the transactions to be tested, and Q is the time for the dependency check. The time complexity is improved to $O(Q+N^2)$, the parallel version of the algorithm. The memory complexity is $O(N+KL)$, where K is the number of transactions in one area handled by one thread and L is the number of threads. The algorithm analysis shows an improvement over similar algorithms, namely the hash table approach [10] and the single matrix approach [11]. In terms of time, the proposed algorithm runs up to four times faster than the compared results, with the worst run being up to three times faster. In terms of memory, the algorithms save 60,000 bytes and 120,000 bytes over the hash table and single matrix approaches, respectively.

The remainder of the paper is structured as follows. Section 2 presents the problem statement. Section 3 gives an overview of related works. In Section 4, the proposed algorithm is described, along with discussion of its complexity analysis. Section 5 presents and discusses the experimental results of the algorithm. Finally, Section 6 presents the conclusion and proposes future enhancements.

## 2. Problem Statement

The problem that this paper targets is as follows. Given a database and a list of transactions performed on this database, the transactions have a total ordering based on the time they were performed. At a certain time, an attack occurs through one or more malicious transactions. However, this attack is not detected immediately, so other non-malicious transactions get affected. Affected transactions have read data written by the

malicious transaction. Thus, not only do malicious transactions initiated by the attack need to be deleted, but every affected transaction should be rolled back as a part of the damage assessment and database recovery processes. Damage assessment can be based on either transactional dependency [12] or data dependency [13].

In transactional dependency, a transaction T1 is dependent on another transaction T2 if T1 reads a data item written by T2. In data dependency, a data item X is dependent on another data item Y if the write operation that is writing X depends on a read operation of Y. The recovery includes rolling back all transactions from the time of the attack, dropping the malicious transactions, and re-executing the affected transactions if they do not conflict with the new state of the database. All these transactions are performed in the same order in which they were originally performed.

As the world is shifting toward online electronic data, databases become more vulnerable to attacks, especially when they contain critical information. With the increasing number of attacks, preventive measures cannot succeed on their own; hence, assessment and recovery approaches become more important. With the large number of records stored in databases, a recovery process may take a significant amount of time. This is unacceptable for certain types of systems, such as healthcare, where availability is a strict requirement. Hence, the research in this field is still ongoing.

Security has been tackled in many research papers. In [14,15], researchers outlined the security problem with malicious providers on the cloud. Other researchers tackled security in IoT systems [16,17]. The authors in [17] worked on how to enable security in IoT systems with the least possible human intervention. In [16], the authors discussed the possibility of increasing the security and reliability of IoT systems. The goal is to find time- and memory-efficient algorithms for detecting affected transactions after detecting a malicious transaction, as well as for database recovery after identifying the affected transactions.

This paper presents a novel approach for damage assessment in a healthcare database after suffering a malicious attack. The proposed algorithm follows the transaction dependency paradigm. Healthcare systems have large databases with highly interlinked data. Thus, if a malicious data item were inserted, for example, all the transactions that read this data item would be affected by the malicious activity. The number of transactions that will be affected depends on how fast the intrusion detection system detects the malicious activity. The damage assessment process should then be executed in the least possible amount of time to identify the affected part of the database. All the affected transactions will be rolled back and re-executed to restore the database to its consistent state (i.e., the state of the database as if malicious activity did not happen). The proposed damage assessment algorithm is inspired by COVID-19 behavior and how it is being controlled. The algorithm is parallelized using multithreading. This proves its efficiency in terms of time and memory consumption, rendering it suitable for healthcare systems where denial of service is very critical and may affect patients' lives.

## 3. Background

Several studies have been published on the topic of database recovery from malicious transactions, aimed at finding efficient algorithms to address the problem stated in Section 2.

### 3.1. Healthcare Information Security and Hash Functions

Cryptographic hash functions are widely used in information security in many areas, like digital signatures and authentication. As an illustration of using hash functions in security, a multi-factor authentication mechanism based on hashing is presented in [3]. Moreover, in [4], an integrated cybersecurity framework for risk management is presented and tested using a power grid system. In healthcare, a framework for remote patient monitoring allowing multiple users from one device is presented in [5]. Another automated method for assessing the impact of privacy and security is presented in [6], based on interdependency graph models and data processing flows.

### 3.2. Classical Methods

Traditional methods in database recovery scan the log file starting from the malicious transaction until the last recorded transaction. In such methods, all changes are rolled back and affected transactions are re-executed [18].

A fusion technique is presented in [19] that follows transaction dependency. The main advantage of this approach is having reduced access to the log file by fusing malicious and affected transactions, undoing any newly fused transaction and redoing it if it was affected and not malicious. Transactions are fused into groups. A set of transactions are in the same group if no malicious or affected transactions occur between them.

The approach presented in [16] is based on utilizing multiple versions of the data to allow executing new read-only transactions while the recovery procedure is running. The proposed solution, called TRACE, proves that there is no downtime while not blocking further read-only operations. In addition, there is a decrease in delays for write transactions. It also has two modes of operation: standby and cleansing.

Although classical methods of database recovery have their advantages, the main disadvantage is the need to scan the whole log file starting from the malicious transaction until the last committed one. Such approaches are time-consuming for large databases that have extremely large log files. Traditional methods do not have a fast way to detect affected transactions. They always must go through affected or benign transactions to roll them back.

### 3.3. Graphs and Agents

Graphs are also used to represent the dependency between the transactions and data items of a database. A graph-based approach is presented in [20] with cold and warm start modes. A cold start mode conducts offline recovery of the database, which increases the downtime. A warm start mode conducts online data recovery with less downtime but possible degradation in performance. Transactions are grouped into malicious and non-malicious types. The data structure used in this approach is a graph where the edges represent dependencies between the nodes of the two groups. This graph structure helps in finding which of the non-malicious transactions are affected by the malicious transactions.

Another approach, presented in [21], uses agents and graphs. Agents receive and forward messages. Graphs are used to represent the dependency between transactions. All messages pass through a single controller agent. Since graphs are not necessarily connected, multiple agents scan multiple graphs for affected transactions and return pointers to a malicious or affected state. This approach can isolate only the affected part of the database.

Graph- and agent-based methods have a disadvantage that is mainly related to graph traversal. The damage assessment process should go back and forth while scanning, which increases the running time of the database recovery as the graph gets bigger.

### 3.4. Clusters and Subclusters

Since log files are usually large, clustering methods split it into segments. Subclustering segments the log file further. The goal of this approach is to deal with smaller log files. To improve the performance of the recovery process, the authors in [12] suggested segmenting the log file by limiting the size of the cluster in one of three possible ways: the first is the number of committed transactions, the second is the memory size of the cluster, and the third is the time window covered by the cluster. There are performance gains of segmentation when using this method.

### 3.5. Matrices

Matrix-based approaches have been studied thoroughly in the literature. They utilize the matrix, represented as a two-dimensional array, as a data structure for storing dependencies between transactions or data items.

A matrix-based approach using data dependency was presented in [22] that utilized a matrix to store the dependencies between transactions. The matrix rows represent the

various transactions, whereas the columns represent some of the data items available in the database. The matrix cells represent whether one transaction has been modified, has been blindly written, or depends on other transactions. In the latter case, an extra array is needed to store the transactions on which the cell depends. The constructed matrix is then used in the recovery process, leading to good running times. However, the disadvantage of this approach is the cost of large memory utilization, especially with the extra array needed.

To address the issue of using an extra array, a new approach was presented in [23] that utilized a single matrix. The matrix dedicates a row for every committed transaction and a column for every data item in the database. The cells store the different values of the modified data items. This approach saves memory because it does not need a complementary data structure, like the case in [22]. However, since the data items are represented as strings, this leads to slower parsing. Additionally, the matrix is generally sparse that consumes time to loop over it.

Haraty and El Sai [10] proposed a solution that utilized a matrix of sorted linked lists. Each linked list represents a committed transaction T and a set of transactions that T depends on (i.e., T reads data from them). Experimental results showed that the algorithm utilized less time and memory in comparison with other approaches.

Another approach in [24] uses a matrix of linked lists based on transactional dependency. This algorithm reserves a linked list for every committed transaction T. A transaction will be connected to the list of T in case it reads data items written by T. Thus, without the need to loop over the matrix, affected transactions are identified. This algorithm outperforms the algorithm in [10] in terms of the running time. However, there is a memory problem; it stores all the committed transactions rather than dependent transactions, leading to high memory usage.

In [25,26], the authors presented a new model aimed at alleviating the drawbacks of [23] related to the parsing problems and increased execution time while dealing with strings. The proposed model used integers to reduce this time. However, its drawback is the sparsity of the dependency matrix. The proposed model deals with this by keeping track of the first non-empty cell and the total number of cells in each row. Thus, it decreases the execution time, but it increases memory consumption. The work in [11] presented a matrix-based approach that utilized a hash table. The hash table has transactions as keys and the transactions that depend on them as list of values to these keys. This decreases the execution time needed to locate all affected transactions from a known malicious or affected transaction. This approach outperformed other matrix-based and agent-based approaches.

Matrix approaches have their performance advantages and are, perhaps, some of the best performing approaches when it comes to database damage assessment and recovery. However, they require significant space in memory, especially with large databases and high numbers of transactions. The sparsity of the matrix is another memory utilization issue that increases the execution time as the matrix gets larger in size.

### 3.6. Column Dependency

A column dependency approach was presented in [27] with two modes: static and online. The static mode entails a downtime where no new transactions are allowed during recovery. The online mode allows new transactions while recovering the database. However, it degrades the performance, since new transactions can be committed and affected while the recovery procedure is running. This requires an additional phase of confinement after the damage assessment and recovery phases.

### 3.7. Application-Specific Recovery

Some researchers presented an application-specific database recovery approach to determine transaction dependencies. They assumed that each application had its own type of dependencies that could be utilized for an enhanced data recovery approach. In [28],

the authors proposed an approach that was specific for banking system applications. Although they have a minimized database recovery time, this approach is not generic and cannot be applied to other types of applications.

Another application-specific approach was presented in [29] for e-government systems. The authors presented an agent-based approach for damage assessment. Agents are assigned to a fixed group of items, and they communicate for more efficient processing. This algorithm is characterized by adding a timestamp to each data item to indicate its version after a write operation. However, each agent must scan the whole log file to perform the damage assessment, which decreases the algorithm efficiency.

### 3.8. Multi-Level Databases

For multi-level databases, [30] proposed a model that was based on transactional dependencies. Their algorithm is based on clustering and graphs to store the dependencies in a multi-level database. The first level represents the dependency matrix, whereas the second one represents the write buckets that are used to track dependencies between transactions. This approach requires scanning the log file once.

## 4. Proposed Method

### 4.1. Assumptions

Some constraints are assumed before the algorithm starts running. These assumptions are as follows:

- There is an intrusion detection system that provides the algorithm by the list of malicious transactions;
- All transactions have a unique sequential ID. Thus, if T2 is a committed transaction, then only T1 is committed before it;
- The log file records the committed transactions and is inaccessible by users;
- The DBMS scheduler history is rigorously serializable (i.e., there is no transaction Ti that will read from transaction Tj if j > i);
- no transaction Ti $\forall$ i = 1, 2, 3, . . . , J starts until a running transaction has ended
- The paper mainly addresses how to find the affected transactions but does not propose a new method for recovery. Therefore, it is also assumed that there is a recovery algorithm that the affected transactions are provided to once found.

### 4.2. Proposed Algorithm

Our proposed algorithm was inspired by the COVID-19 spread behavior and how it is being controlled. In the case of COVID-19 infection, there is a person with COVID-19 P that tests positive and should be isolated. Additionally, any person I who gets in touch with P should also be isolated and prevented from further contact with others until he or she tests positive or negative. If I is positive, then any person who got in touch with I should also be isolated and tested. This procedure is repeated until no further positive cases are discovered.

Similarly, in databases, there is a malicious transaction that starts the infection. Transactions interact with each other through reading and writing data items. Hence, when a transaction reads a data item written by a malicious transaction, it will get affected. Thus, all the transactions that could have been affected by the malicious transaction should be isolated and will be added to the testing list. This procedure is repeated for each transaction in the testing list until the list is empty. Each test on a transaction will produce a list, possibly empty, of affected transactions, thus keeping the algorithm going until all affected transactions have been found. To increase the efficiency of the algorithm by decreasing the time and memory requirements, threads are used to execute in parallel. When the intrusion detection system detects a malicious transaction with an ID m, the algorithm starts execution. Threads will start executing in parallel to check whether any committed transaction reads a data item written by m. If the test is positive, this transaction will be added to the testing list. Once again, the threads will execute simultaneously on

every transaction ID in the testing list to check for more infected transactions. This process continues until all the transactions in the testing list are tested.

### 4.3. Pseudocode

The pseudo code of the algorithm is as follows, and a flowchart follows in Figure 1.

---

**Algorithm 1: Pseudocode**

---

*testing_list* ← list of malicious transaction(s)
*threads* <- list of threads covering the various areas
Each thread will run the below steps:
*i* ← 0
while not all threads are waiting on *testing_list* do
if *i* < size(*testing_list*) then
*current_transaction* ← *testing_list* [*i*]
if *current_transaction* ∈ area of thread then
find transactions in area that interacted with    *current_transaction*
append these transactions to *testing_list* if not already in it
else
skip current transaction since out of thread's area
endif
*i* ← *i* + 1
else
wait on testing_list for new elements
endif
endwhile

---

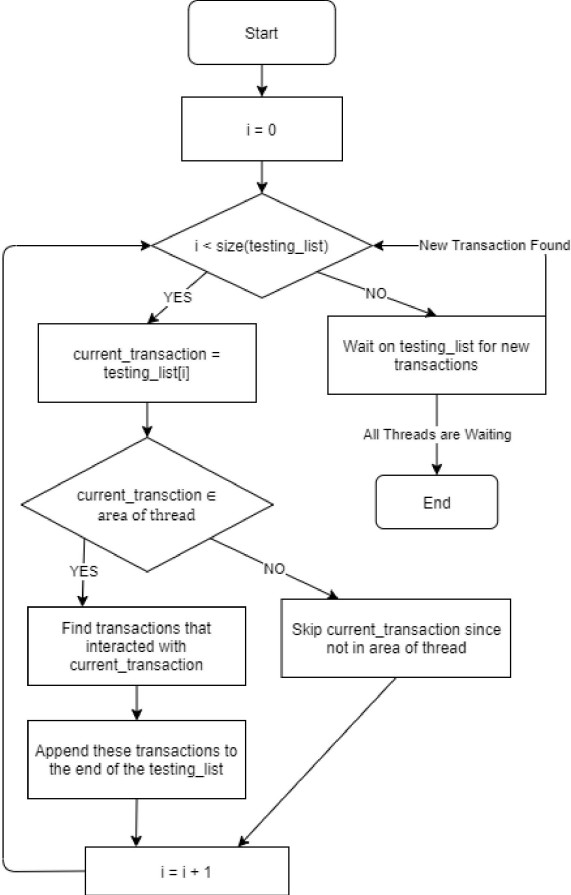

**Figure 1.** A flowchart for the above algorithm.

### 4.4. Example

Consider the following scenario of transactions. We have six transactions T1–T6 that are reading and writing data. Each thread is responsible for an area that contains a group of transactions. The areas could be, for this example, the relationships between the tables or simply all pair-wise combinations of tables:

- T1 reads A to write C;
- T2 reads C to write B;
- T3 reads C;
- T4 reads B and C to write D;
- T5 writes E blindly;
- T6 reads B and E to write F.

Assume that T1 is the malicious transaction. T1 writes C. Thus, any transaction that reads C will be infected:

*testing_list*: [T1]

All threads start to test from T1, which is in the testing list, to add transactions from their areas that interacted with T1 (i.e., read data item C). They will find T2, T3, and T4:

*testing_list*: [T1, T2, T3, T4]

Now, each thread continues from its last checked index, so, they check for T2. They find the transactions that interacted with T2 (i.e., read data item B), which are T4 and T6:

*testing_list*: [T1, T2, T3, T4, T6]

T3 writes E blindly, which means that it does not read any data item, so it will not update the testing list.

The next transaction is T4, and when the threads run, they return transactions that read D, which is an empty list since no transaction read D. Thus, the testing list remains unchanged.

The threads then test T6, but do not add anything to the testing list. Hence, the final list of transactions that are affected is as follows:

affected transactions: [T2, T3, T4, T6]

All the threads are running in parallel. Each thread can read the index of the last tested transaction in the testing list so that it starts from the next transaction. A transaction may belong to some areas and not others, so the thread moves through the testing list at different speeds. The testing list is synchronized to ensure proper appending and no duplicates. Once all threads are at the end of the testing list, then there are no more affected transactions, and termination is initiated.

### 4.5. Complexity Analysis

Each thread handles a specific area. As an optimization and not as an essential part of the algorithm, each thread will first query the set of transactions it could be responsible for and stores them in an in-memory set. Before handling any transaction or checking whether this transaction has affected another one, the thread first checks if the transaction belongs to its set. This saves unnecessary testing. The time needed to fill this set is the time needed to perform the query on the database or log file and loop over the retrieved transactions. The time needed to insert into the set and test its belonging to the set is the constant O(1), as per the Java HashSet documentation, and may fall back to a TreeSet O(log n) in the case of many collisions. The memory consumed by this set of transactions is equal to the number of transactions the thread is responsible for because the transactions are stored according to transaction IDs, which are integers.

Note that the above set of transactions per thread area can be built during the stage of transaction processing in the database or while the damage assessment algorithm is running. In this paper, it is built during the damage assessment algorithm to give a better overview of the algorithm's overall time.

The list of affected transaction IDs is represented as an array list that has the time complexity of amortized O(1). The transaction ID at a given index can also be retrieved in O(1). The maximum size of this list is equal to the total number of affected transactions for

the given set of malicious transactions. It is ensured that the list does not have duplicates, so the algorithm loops over the list before the insertion to check whether the transaction exists or not. This makes the complexity of insertion into the list O(n), where n is the size of the list. Being a simple way to ensure no duplicate transactions in the testing list, this approach is used in our implementation. Another solution to prevent duplicates is working on the thread area to remove a transaction ID once this transaction has been processed for the first time. This will prevent processing the same transaction twice. Thus, the final list can then be cleared of duplicates, if any.

The threads will check the first unchecked transaction ID in the testing list, based on a last-checked index they have. Once they have a transaction, they test its belonging to their set to know if they should go further or continue to the next transaction. If they can handle the given transaction ID, they perform an SQL query by joining the tables on the relationship in a query based on (PK-FK), where PK is the primary key and FK is the foreign key, and by selecting the transaction IDs in the referencing rows based on the given affected transaction ID in the referenced row or any other method of checking for dependency based on the area of the thread. The retrieved transactions are then added to the affected list of transactions to be checked by all threads.

To prevent race conditions and duplicates while inserting and retrieving IDs from the list, such operations are synchronized by using a wrapper class for the list with synchronized methods. Once a thread reaches the end of the list, it stops and waits for new affected transactions to test. The main thread of the program runs a routine check on whether all threads are waiting or not. If all threads are waiting, then all present transactions have been checked by all threads, and no new affected transactions can be found. This is the condition for stopping, so the main thread can now terminate, join the threads, and retrieve the final affected transactions list.

*4.6. Memory Complexity*

Let the number of affected and malicious transactions be N. Let the maximum number of transactions in one area handled by one thread be K. Let the number of areas, which is the same number of threads, be L.

The memory complexity of the algorithm is O(N) if we do not use the optimization of storing sets in threads, since the isolation queue will have at most N elements. If the optimization is used, the memory complexity is O(N+KL) because each thread will store a set of transactions that it can handle. Generally, this is expected to be close to O(N+M). Note that this is linear in the number of transactions, unlike in other algorithms that require quadratic space such as O(M2) or O(MN). As such, we expect our algorithm to be more memory efficient than other approaches with two-dimensional data structures.

*4.7. Time Complexity*

The time complexity of the algorithm is a bit challenging to compute, since it depends on how the affected transactions are distributed among the different areas and whether many threads will be working in parallel or a few threads will handle all the tests.

Assume a database query, or any test of dependency between transactions, takes O(Q) time. In total, there will be N insertions into the testing list, each insertion taking O(N) to check if the transaction is already present before inserting it. Each thread will first get the transactions that it can handle in its area and insert them into its set in O(Q+K) time. Each thread will poll the list N times, with each 400th time taking O(1) because the index to be retrieved is known. If the transaction is not in the threads area, the thread will simply move on. Checking whether the transaction is in the area requires O(1) time for the set. If the transaction is in the area, the thread performs a query O(Q), looping over the affected transactions O(N) and inserting them into the testing list O(N). This can take O(Q + N2) time per affected transaction. Note that the actual time is much less, since we considered that a thread may find all N affected transactions in one query and still proceed to find transactions, whereas this is highly unlikely to happen.

Considering a serial algorithm with only one thread, the thread will have to handle N affected transactions, and K would be M, so the total time complexity would be O(Q +M + N(Q + N2)) = O(M + NQ + N3). The best-case scenario is when the affected transactions are distributed evenly among the various areas of the threads. With L threads, the number of transactions handled by a single thread would be M/L, and ideally, each thread would handle N/L affected transactions, so the time complexity would be O((M + NQ + N3)/L). This shows how scalable the algorithm is when L increases.

*4.8. Comparison*

Table 1 shows the comparison of time and memory complexity between three algorithms: our algorithm, Boukhari's (2020), and Haraty and Sai's (2017). Based on the table, our algorithm is expected to consume far less memory than the other algorithms, since it is not quadratic in the number of transactions. This is especially noticeable when the number of transactions to be processed is large, as would be the case with real-life databases. Regarding time complexity, as the number of areas and threads increases, our algorithm becomes more efficient, and it is expected to outperform the other algorithms with an appropriate number of threads.

**Table 1.** Comparison of the three algorithms considered in the experiments. The time complexity does not account for the time to build the needed data structures. M = total number of transactions; N = number of malicious and affected transactions; R = number of transactions that the transactions depend on; L = number of threads or areas; K = number of transactions handled by one thread or in one area; and Q = time to check dependency between transactions.

|  | Time Complexity | Memory Complexity | Needs Existing Data Structure |
|---|---|---|---|
| Our Algorithm | O((M+NQ+N3)/L) | O(N+KL) | No |
| Boukhari's (2020) | O(M) | O(RM) | Yes |
| Haraty and Sai's (2017) | O(MR) | O(RM) | Yes |

## 5. Experimental Results

The algorithm was implemented to exploit the multi-threading feature in the Java programming language. A thread is a lightweight subprocess which is the smallest unit of processing. This feature permits the execution of two or more threads simultaneously that execute parts of the program for the highest utilization of CPU.

To verify the efficiency of the algorithm, experiments were conducted to compare it with similar algorithms presented in [10,11]. These two algorithms were specifically chosen since they were very efficient and perhaps had the best time and memory performance among many algorithms. Note that the two algorithms being compared to the new algorithm needed a data structure that was built along with the database transactions and log file. This added an implicit time to the algorithms, which in our case was considered in the algorithms' running times. The transactions, tables, data, and other factors used in the below tests were the ones used in the papers, specifically the Northwind database [31].

The Northwind database is a sample database used for demonstration purposes of some of Microsoft's products, especially when it comes to SQL Server. It has been used for experiments in the field of damage assessment and database recovery. The data in the database belongs to a made-up company named Northwind Traders and includes information about sales.

A particular version of the Northwind database, written for the MySQL syntax, was used. The SQL file was loaded before every test. From the database, only transactions on the below tables were studied in the experiments:

- Employees: the employees of the company;
- Suppliers: the suppliers of products;
- Customers: the customers of the company;
- Categories: the categories of the products;

- Products: available products;
- Orders: orders placed;
- OrderDetails: details about the order.

The studied relationships between these tables constituting the different areas were as follows:

- Categories and Products;
- Suppliers and Products;
- Customers and Orders;
- Products and OrderDetails;
- Orders and OrderDetails.

The transactions considered were INSERT transactions, where each transaction inserted a row into a table. In other words, the above seven tables were empty at the beginning, and they were filled completely, one row at a time, with such transactions. Thus, the dependency between transaction arose from the primary key (PK) and foreign key (FK) relationships between the different tables. The threads were assigned to areas by the proposed multi-threading algorithm. These areas were the relationships between the tables. In this case, there were five relationships and thus five threads. Each thread was responsible for checking whether a given transaction belonged to its area or not. If it belonged, then the thread queried the affected transactions that referenced it and added them to the list of affected transactions.

In every run of the algorithm, the SQL file defining the database was loaded, and the log file was generated. The log file contained the transactions listed in the chronological order in which they were performed.

### 5.1. Experimental Setup

All algorithms tested below were implemented using the Java programming language, specifically version 1.8.0_45. For connection to the database, the mysql.mysql-connector-java library (version 8.0.21) was used. For measuring the memory consumption of specific objects, the com.carrotsearch.java-sizeof library (version 0.0.5) was used. The code was packaged along with all dependencies using Apache Maven version 3.6.1 into a jar file, which was then run on the different algorithms and with different malicious transaction IDs.

The environment of simulation had a Windows 10 operating system with an Intel®Core™ i7- 4700 MQ 2.40 GHz CPU with 4 cores and 16 GB of RAM. For the database, XAMPP was used to run a MySQL (version 15.1 distribution 10.4.13-MariaDB) DBMS on which the database was loaded before running each test. The different malicious transaction IDs tested were as follows: 1, 20, 50, 100, 150, 200, 400, 600, 800, and 1000. Since the time taken by the algorithm may have differed due to the load on the system, the response time of the database, and cores available, among other factors, all the algorithms ran 10 times for each malicious transaction ID. Among those runs, the best, worst, and average time were recorded. The memory required was generally not affected by other factors and hence had one reporting, which was the same for all runs.

The number 10 was chosen to ensure that other factors mentioned above, such as other processes on the system, availability of cores, or database response time, did not greatly affect the reported results or provide a disadvantage to some algorithm. We understand that when measuring small units such as microseconds, the results may vary between runs, so we assume the average time reported to be the most accurate comparison. Going beyond 10 runs did not considerably change the reported results.

### 5.2. Results and Discussions

To get a better idea of the number of affected transactions for each malicious transaction ID, Figure 2 is provided.

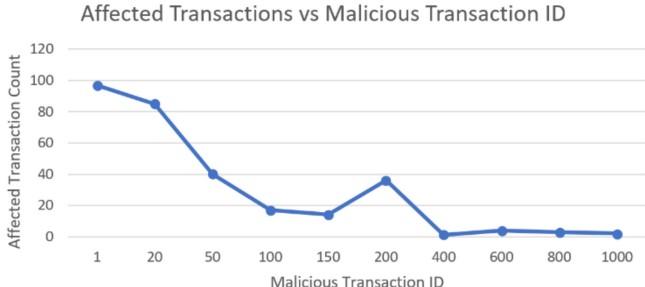

**Figure 2.** Number of affected transactions as a function of the malicious transaction ID.

In general, the number of affected transactions tended to decrease as the malicious transaction ID increased and got closer to the last recorded transaction. It is interesting, however, that the number rose at malicious transaction ID 200, since many of the rows inserted later referenced the row inserted by this transaction. It is also important to note that these rows belonged to a few tables, so the affected transactions were not diverse among the different tables with large malicious transaction IDs (like 200), as was the case with small malicious transaction IDs, such as 1 or 20.

*5.3. Time*

In Figure 3, Figure 4, Figure 5, a comparison of the best, worst, and average times, respectively, of 10 runs for each of the three algorithms over the studied malicious transaction IDs are shown. All times reported are in microseconds.

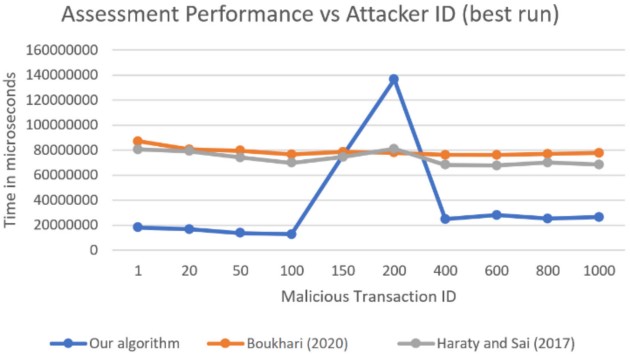

**Figure 3.** Comparison of the best times of three algorithms as a function of the malicious transaction ID.

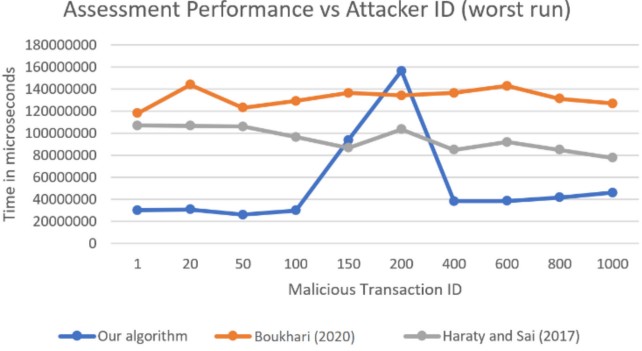

**Figure 4.** Comparison of the worst times of three algorithms as a function of the malicious transaction ID.

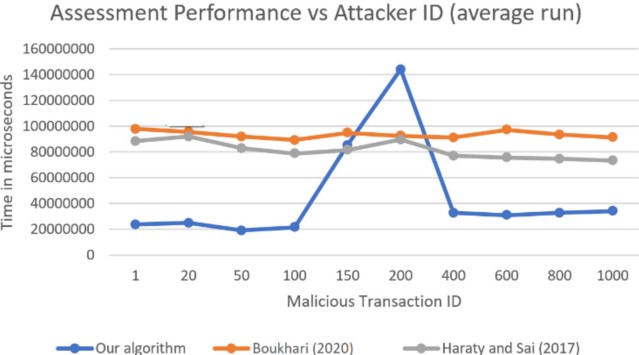

**Figure 5.** Comparison of the average times of three algorithms as a function of the malicious transaction ID.

The proposed algorithm was faster than the other two algorithms (presented in [10] and [11]) for almost all transactions except for two: 150 and 200. It is clear in Figure 5 that in the average of all 10 runs for attacker ID 1, our algorithm was almost five times faster than the Boukhari approach [11] and more than four times faster than the Haraty and Sai [10] approach. It was also four times faster than the other algorithms for malicious transaction IDs 20, 50, and 100. For transaction IDs 400, 600, 800, and 1000, our algorithm was more than twice as fast as the other algorithms. For transaction ID 150, our algorithm had almost the same running time as in [10] and could be still considered competitive. The only problem was transaction ID 200, which will be explained later in this section.

In the best run, as shown in Figure 3, the results seemed similar to those of the average case, with our algorithm being four times faster in the first few attacker IDs and more than two times faster in the last attacker IDs. At 150, it had similar times than the other algorithms, with 200 remaining a standout point. In the worst recorded run shown in Figure 4, our algorithm was almost twice as fast as the algorithm in [10] and almost three times as fast as the algorithm in [11]. Note that when the worst runs of all the algorithms were recorded, the problem at attacker ID 200 seemed to be reduced, and the recorded value seemed normal with respect to the other algorithms.

First, the reason the algorithm was faster in general was because it utilized multi-threading and thus could check and locate affected transactions faster due to parallelism. It tended to be faster on the first few attacker IDs since the number of affected transactions was larger, meaning that it could finish more work in parallel while other algorithms must do it serially. In addition, when there was an early transaction where the transaction ID was small, it was likely that the transactions it affected belonged to multiple areas, which led to better parallelism and better performance. For malicious transaction ID 150, the running time was like the other two algorithms, specifically [10], and generally, our algorithm performed well compared with theirs. The problematic results were for malicious transaction ID 200, and this was due to what was fleetingly explained in a previous section regarding the number of affected transactions being high for malicious transaction 200 and how the affected transaction belonged to very few tables (areas), making the algorithm run serially.

We looked more into the results and found that malicious transaction IDs 150 and 200 specifically affected only one relationship, which was that between the Order and OrderDetails tables. Therefore, there was only one area out of the six to query, which meant that only one thread was doing all the work. Since one thread must do the querying for each transaction, our approach became much slower because its main strength was parallelism. The other two algorithms make use of a data structure that is built at the beginning of the damage assessment process. Therefore, they do not rely on database queries later. Thus, they do not get affected by the number of affected transactions as much as our algorithm does, or by whether these transactions affect a single table or relationship. Specifically, when the malicious transaction ID was 200, our algorithm was using one

thread to query and find 36 affected transactions, and this was what caused the spike in the graphs. However, even with that, it was not much worse than the other two algorithms, especially when it came to the worst recorded run out of the 10 runs.

This drawback can be improved by using another mode of parallelization. For example, the threads can be allowed to query more than one area, which alleviates the worst case when only one area is affected. Moreover, the proposed multi-threaded algorithm does not make use of a specialized data structure that comes pre-filled, as is the case with the other two algorithms. If it does, then it would certainly overcome this worst case. Assume that it stores a matrix or hash table of dependencies between transactions. Then, it does not need to query the database and can rely on the information in the data structure. By using this hybrid approach, when multiple areas are affected, our algorithm will always perform better due to parallelism. When only one area is affected, it falls back to one of the other two serial algorithms.

### 5.4. Memory

Figure 6 shows a comparison of the average of 10 runs for each of the three algorithms over the studied malicious transaction IDs. Note that the memory did not change between the runs, since it did not depend on variables such as the CPU, 550 database response time, or available cores, so the reported average was also the recorded memory for each of the 10 runs. All memory results are reported in bytes.

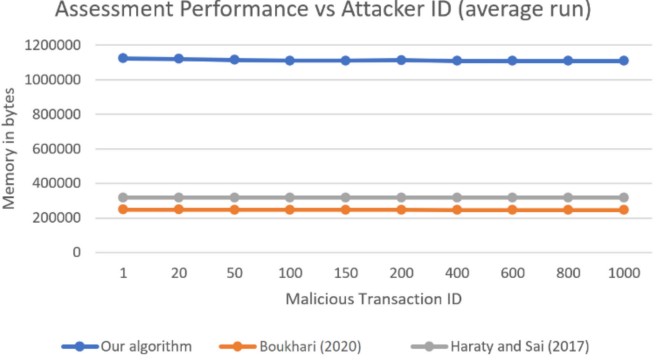

**Figure 6.** Comparison of the total memory used in the average of 10 runs by three algorithms as a function of the malicious transaction ID.

The figure shows surprising results, where our algorithm consistently consumed 3–4 times as much memory as the other two algorithms. It seems that our algorithm was performing very poorly from the memory perspective, which should not be the case. As mentioned previously, the other two algorithms use two-dimensional data structures to keep track of dependencies between transactions, which means quadratic memory. On the other hand, our algorithm uses only one-dimensional data structures that hold the list of affected transactions (560) and the set of transactions in each area. These should consume less memory than those in the other two algorithms.

The reason for the discrepancy between what was expected and what was recorded is that Java threads consume additional memory. Each of the six threads that we had in our algorithm was consuming some extra memory of its own just for being a thread. Whenever a thread is created, Java will allocate a space in the memory for this thread, namely some stack size and possible some data structures and variables. The stack size can be reduced to some minimum value (108k in our case), but that depends on the system being used (sometimes the value is 512k, for example). This means that there will always be memory consumed by threads, and that increases the memory used by our algorithm. Figure 7 shows a comparison between the memory used by the other two algorithms and our algorithm when there was only one thread.

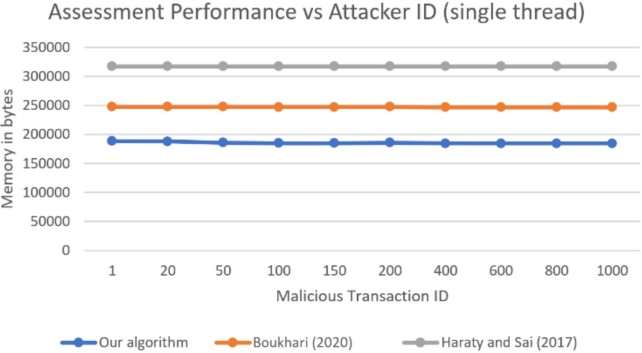

**Figure 7.** Comparison of the total memory used by three single-threaded algorithms as a function of the malicious transaction ID.

The figure clearly shows how the data stored by our algorithm, in addition to the memory of a whole thread, was less than the memory used by the other two algorithms, considering the data used and a single thread. From this graph, our algorithm consumed about 60,000 less bytes than the hash table approach and about 120,000 less bytes than the single matrix approach. These results prove that when our algorithm runs with one thread, it consumes less memory than other algorithms. To give a better idea of how the memory compares with other algorithms regardless of thread memory, we ran the tests one more time, only measuring the memory used by data structures and stored variables. Instead of taking the total assessment program memory, as in Figure 6, or one-thread assessment memory, as in Figure 7, we only measured the program data size, namely the data structures used, whether hash table, matrix, lists, sets, or any other structure. The results are shown in Figure 8.

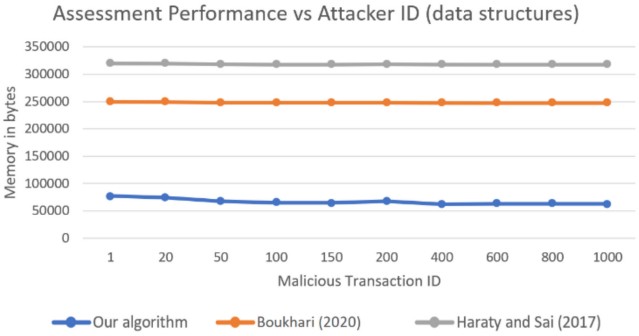

**Figure 8.** Comparison of the data structure memories (ignoring thread overhead memory) used by three algorithms as a function of the malicious transaction ID.

As can be seen from Figure 8, our algorithm consumed very little memory compared with the other two algorithms once the extra memory allocated for threads was ignored. This shows how threads consume a lot of memory with respect to the actual algorithm data being stored. When only the algorithm data was considered, our algorithm consumed about 3.5 times less memory than the algorithm in [11] and about 4.5 times less memory than that in [10]. The above extra memory used by the threads can be considered constant overhead. Therefore, what really matters when scaling to large databases is the memory consumed by actual data stored by the algorithm. This means that when the number of transactions gets much larger, the case will be that our algorithm consumes less overall memory—including threads—than the other two algorithms.

### 5.5. Interpretation

The above results verify our calculations of complexity presented in previous sections. Our algorithm did in fact take less time and memory in general. There were, however,

cases in which our algorithm performed worse, such as when only very few areas were affected by the malicious transaction or when the number of affected transactions in a single area was much larger than the rest. The linear memory used was also verified in the experiments, although the algorithm performed worse from the memory perspective due to the extra thread memory.

Our algorithm primarily targets large databases that have huge numbers of transactions. With such scenarios, the parallelism gained from threads would be much more evident, making our algorithm much faster than other approaches. In addition, when databases are large with many tables and transactions, it is less likely to be caught in the worst case of having many affected transactions grouped into very few areas. As for memory, the memory consumed by threads tends to be negligible when compared with the memory consumed by algorithm data structures when huge numbers of transactions are involved.

With lots of transactions, our algorithm needs linear memory, because the extra thread memory can be assumed to be constant. Hence, it will need much less memory than other approaches that require quadratic memory, which would be catastrophic with millions of transactions, for example. Finally, our algorithm does not need a data structure to be built along the way, and this saves time on regular database operations.

## 6. Conclusion and Future Work

Healthcare databases are a prized target for attackers because they enable identity theft and cybercrimes. Any denial of service of healthcare databases affects patients' lives.

The gravest metric to be considered while securing healthcare databases is pace. Patient history should be recovered as soon as possible, and healthcare transactions should not be affected for a long time by an attack. Hence, database assessment for the healthcare system recovery should be effective to prevent extended stoppages while enacting these measures.

This paper presented a new algorithm for database damage assessment after a malicious attack on a healthcare system. We designed an algorithm for damage assessment that can be used after a malicious attack while trying to restore the database into its consistent state. The proposed algorithm was inspired by COVID-19 spread behavior. The malicious transaction was viewed as a COVID19 virus. Then, we dealt with the infected transactions as if they were people with COVID-19. Any transaction that interacted with an infected one was isolated. The algorithm uses multiple threads that simultaneously search several areas to locate affected transactions in the least possible time. Experiments proved that this approach meets the performance expectations and outperforms the two existing efficient algorithms. On average, our algorithm is five times faster than other existing ones. In addition, our algorithm is more efficient in terms of memory. Since the attacks on healthcare systems require fast recovery to decrease downtime, and healthcare databases are usually large in size, such an algorithm can be utilized.

As an extension of this work, transactions can be further studied to enhance the algorithm. Blind writes are transactions that write the data item without reading it. Thus, an infected data item will be refreshed after a blind write. Considering such types of transactions decreases the number of transactions to be tested, which will decrease the damage assessment time. Moreover, the transactions' natures can be studied to decrease unnecessary dependencies and thus decrease the damage assessment time. In healthcare databases, for instance, a patient cannot depend on another patient since they are separate entities. However, a patient will always depend on a doctor; thus, a malicious transaction that includes a doctor should lead to a doctor automatically moving all their patients to the affected data list without checking any constraints. This step will greatly decrease the damage assessment time.

**Author Contributions:** Conceptualization, S.K.; software, K.A.K.; formal analysis, R.A.H.; supervision, O.A. All authors have read and agreed to the published version of the manuscript

**Funding:** This research received no external funding.

**Data Availability Statement:** Not applicable, the study does not report any data.

**Conflicts of Interest:** The authors declare no conflict of interest.

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
