# Peer review of "A Parallelized Database Damage Assessment Approach after Cyberattack for Healthcare Systems"

_futureinternet, doi:10.3390/fi13040090_

Round 1

Reviewer 1 Report

I have reviewed the manuscript with ID futureinternet-1146019, intitle “A Parallelized Database Damage Assessment Approach After Cyberattack for Healthcare Systems ”, which may be of interest to the community of this Journal, however, I have some concerns that I list below:
1. Abstract.  Author should follow the style of a structured abstract, which is based on the IMRAD structure of a paper, but without using headings . In other words, give a background and motivation to the paper, a brief description of the methods, the principle results, then conclusions or interpretations.  In the particular case of this article, it is necessary to clarify the method,  principle results, conclusions or interpretations.
2. Introduction. The references (quotes) must be enumerated in order, i.e., as they are cited. This numbering in ascending order must be in the entire manuscript.
3.  Introduction.  The introduction is poor. The function of the Introduction is to: Establish the context of the work being reported. This is accomplished by discussing the relevant primary research literature (with citations) and summarizing our current understanding of the problem you are investigating;  (Relevant works are not being cited in this Introduction, which gives the impression that they are being ignored.)
- State the purpose of the work in the form of the hypothesis, question, or problem you investigated; and, 
- Briefly explain your rationale and approach and, whenever possible, the possible outcomes your study can reveal. 
- Therefore, I suggest that the Introduction be rewritten, so that the manuscript sounds more interesting for the scientific community. Please rewrite a more complete Introduction following the above described, it is important to thoroughly review the literature and state of the art (please review recent papers, 2018-2021), and cite the most relevant related work.
- Introduction. Typically in information security, Cryptographic Hash Functions are used to detect alterations (data corruption) in the information, I consider that in this manuscript it is necessary to cite some relevant works (recents 2018-2021) in this regard and it would be interesting if the authors compare results of the performance of the proposed algorithm versus the well-known Hash Functions. 
4. State of the art. The authors did not correctly review the literature and are ignoring some relevant papers that can be easily found in this Journal and this Editorial. Which is important to cite and briefly describe the state of the art papers and mention the disadvantages, trends and open problems.
5. Section 4 THE MODE, I suggest it be renamed "Proposed Method”, because Model, normally in scientific research represents a Mathematical Model, and in this manuscript it is not the case.
6. Section 4. The method must be described in more detail, because it is not clear how the algorithm detects the "contagion" or "virus" in the database? 
7. So it is suggested that the authors clarify in the algorithm what happens when the virus is detected? how long does the quarantine last? How is the correction or healing performed? etc.
8. In addition, to clarify the method, it is suggested to add a detailed flow chart that describes the operation of the proposed algorithm.
9.  The authors mention that the algorithm works in a similar way to the dynamics of Covid-19, however, in the particular case of this algorithm, the state of quarantine or isolation is not clear.
10. Experimental Setup. Please specify the version of the Java language used.
11. The quality of all the figures must be improved, they look blurry, it seems like screenshoot.
12. Line 437, the authors say…. The proposed algorithm is faster than the other two algorithms [please put the references]
13. To verify the dataset and the performance of the proposed algorithm, and with the purpose that other readers and authors can verify and compare the performance of new algorithms, it is suggested that the authors share their dataset as supplementary data, according to the Open Access policies of this publisher.
14. In addition, the data set should be described in more detail in the manuscript.
15. The results presented in Figures 1-7, it is suggested that the authors analyze, discuss and interpret in greater detail.
16. It is suggested to add a comparative table of characteristics (features) of the proposed algorithm versus the related work, please analyze and interpret the information shown in the table.
17. The conclusions should be updated according to the new results and should emphasize the main results.
18. References. It is recommended to homogenize the reference fields. For better visibility, please add the  DOI (when apply) to all References. Another references are missing important fields, as Vol. or Issue, etc. It is important to present the references clearly and with all the required fields. 
19. After reviewing the manuscript, I consider that the contribution to the state of the art is still not clear, therefore, I suggest that the authors add another metric to assess its performance of the proposed algorithm, for example, they can use Halstead Complexity Analysis (Halstead's metrics) and compare the results versus the related work or another equivalent metric to evaluate algorithms.
I hope these comments and suggestions help to improve the quality of the manuscript and clarify the contribution to the state of the art. In case that the authors decide to resubmit the manuscript, please highlight all changes in another color in the revised version of the manuscript.

Author Response

Reviewer 1:

I have reviewed the manuscript with ID futureinternet-1146019, in title “A Parallelized Database Damage Assessment Approach After Cyberattack for Healthcare Systems”, which may be of interest to the community of this Journal, however, I have some concerns that I list below:

  1. Abstract. Author should follow the style of a structured abstract, which is based on the IMRAD structure of a paper, but without using headings. In other words, give a background and motivation to the paper, a brief description of the methods, the principal results, then conclusions or interpretations.  In the case of this article, it is necessary to clarify the method, principal results, conclusions, or interpretations.

The abstract has been updated to include a brief description of the method along with a summary of results and conclusion. The abstract is copied below.

Abstract--In the current Internet of Things era, all companies shifted from paper-based data to electronic format. Although this shift increased the efficiency of data processing, yet it has security drawbacks. Healthcare databases are a precious target for attackers because they facilitate identity theft and cybercrimes. Basically, healthcare data should be readily available to authorized healthcare professionals. Any denial of service for such databases could affect patients’ lives. Although many security preventive measures are deployed in all healthcare sectors, statistics prove that there is always a successful attack. Thus, the demand for defensive information warfare paradigms increases. The most critical metric to be considered, in any of these paradigms, is the efficiency in damage assessment. Healthcare transactions should not be affected for a long time by an attack. This paper presents an approach for database damage assessment for healthcare systems. Inspired by the current behavior of COVID-19 infections, our approach views the damage assessment problem in the same way. The malicious transactions will be viewed as if they are COVID-19 viruses. The transactions will be treated as if they are people infected COVID-19. Other transactions will be isolated to prevent them from taking the infection onward. The challenge of this research is to discover the infected transactions in a minimal time.  The proposed parallel algorithm is based on the transaction dependency paradigm, with a time complexity O(Q+N^2), (N: number of insertions in testing list, Q: time for dependency check). The memory complexity of the algorithm is O(N+KL), (N: number of malicious transactions, K: transaction in one area handled by one thread, L: number of threads). Since the damage assessment time is directly proportional to the denial-of-service time, the proposed algorithm provides a minimized execution time. Our algorithm is a novel approach that outperforms other existing algorithms in this domain, in terms of both time and memory, with up to 4 times speed up in terms of time, and 120,000 bytes less in terms of memory.

  1. Introduction. The references (quotes) must be enumerated in order, i.e., as they are cited. This numbering in ascending order must be in the entire manuscript.

The references have been updated. References that are nor cited (1, 9, 12, 13, 27 in the old manuscript) have been removed, and numbering of all references has been updated to be in ascending order. The following references have been added:

Anderson, J. M.; Why we need a new definition of information security. Computers & Security. 22 (4): 2003, pp. 308–313. doi:10.1016/S0167-4048(03)00407-3.

Ahmed, A.A.; Ahmed, W.A. An Effective Multifactor Authentication Mechanism Based on Combiners of Hash Function over Internet of Things. Sensors 2019, 19, 3663. https://doi.org/10.3390/s19173663

Kure, H.I.; Islam, S.; Razzaque, M.A. An Integrated Cyber Security Risk Management Approach for a Cyber-Physical System. Appl. Sci. 2018, 8, 898. https://doi.org/10.3390/app8060898

Ondiege, B.; Clarke, M.; Mapp, G. Exploring a New Security Framework for Remote Patient Monitoring Devices. Computers 2017, 6, 11. https://doi.org/10.3390/computers6010011

Papamartzivanos, D.; Menesidou, S.A.; Gouvas, P.; Giannetsos, T. A Perfect Match: Converging and Automating Privacy and Security Impact Assessment On-the-Fly. Future Internet 2021, 13, 30. https://doi.org/10.3390/fi13020030

References are copied below.

REFERENCES

  1. Stapleton, J.J.: Security without Obscurity: A Guide to Confidentiality, Authentication, and Integrity (1st ed.). Auerbach Publications, 2014, pp. 355. DOI:https://doi.org/10.1201/b16885.
  2. Anderson, J. M.; Why we need a new definition of information security. Computers & Security. 22 (4): 2003, pp. 308–313. doi:1016/S0167-4048(03)00407-3.
  3. Ahmed, A.A.; Ahmed, W.A. An Effective Multifactor Authentication Mechanism Based on Combiners of Hash Function over Internet of Things. Sensors 2019, 19, 3663. https://doi.org/10.3390/s19173663.
  4. Kure, H.I.; Islam, S.; Razzaque, M.A. An Integrated Cyber Security Risk Management Approach for a Cyber-Physical System. Sci. 2018, 8, 898. https://doi.org/10.3390/app8060898
  5. Ondiege, B.; Clarke, M.; Mapp, G. Exploring a New Security Framework for Remote Patient Monitoring Devices. Computers 2017, 6, 11. https://doi.org/10.3390/computers6010011
  6. Papamartzivanos, D.; Menesidou, S.A.; Gouvas, P.; Giannetsos, T. A Perfect Match: Converging and Automating Privacy and Security Impact Assessment On-the-Fly. Future Internet 2021, 13, 30. https://doi.org/10.3390/fi13020030
  7. Hutchinson, B., Hutchinson, W., Warren, M.: Information Warfare: Corporate Attack and Defence in a Digital World. 1st edn. Butterworth-Heinemann, 2001.
  8. Gao, Y., Y. Peng, Feng Xie, Wei Zhao, D. Wang, Xuefeng Han, Tianbo Lu and Z. Li: Analysis of Security Threats and Vulnerability for Cyber-Physical Systems. Proceedings of 2013 3rd International Conference on Computer Science and Network Technology, 2013, pp. 50-55. DOI:https://doi.org/10.1109/ICCSNT.2013.6967062.
  9. Smith, T. T.: Examining Data Privacy Breaches in Healthcare. Walden Dissertations and Doctoral Studies, 2016, 2623. https://scholarworks.waldenu.edu/dissertations/2623.
  10. Ragothaman P., Panda B.: Analyzing Transaction Logs for Effective Damage Assessment. In: Gudes E., Shenoi S. (eds) Research Directions in Data and Applications Security. IFIP — The International Federation for Information Processing, vol 128. Springer, Boston, MA, 2003. DOI:https://doi.org/10.1007/978-0-387-35697-6_8.
  11. Panda, B., Haque, K.A.: Extended Data Dependency Approach: A Robust Way of Rebuilding Database. In Proceedings of the 2002 ACM Symposium on Applied Computing (SAC '02). Association for Computing Machinery, New York, NY, USA, 2002, pp. 446–452. DOI:https://doi.org/10.1145/508791.508875
  12. Dbouk, T., Mourad, A., Otrok, H., Tout, H. and Talhi, C.: A Novel Ad-Hoc Mobile Edge Cloud Offering Security Services Through Intelligent Resource-Aware Offloading. IEEE Transactions on Network and Service Management 16(4), 2019, pp. 1665–1680. DOI:10.1109/TNSM.2019.2939221.
  13. Dbouk, T., Mourad, A., Otrok, H., Talhi, C.: Towards ad-hoc cloud-based approach for mobile intrusion detection. In: 12th International Conference on Wireless and Mobile Computing, Networking and Communications (WiMob), IEEE, Los Alamitos, CA, USA, 2016, pp. 1–8. DOI:http://dx.doi.org/10.1109/WiMOB.2016.7763251.
  14. Kherraf, N., Sharafeddine, S., Assi, C.M. and Ghrayeb, A.: Latency and Reliability Aware Workload Assignment in IoT Networks with Mobile Edge Clouds. IEEE Transactions on Network and Service Management 16(4), 2019, pp. 1435–1449. DOI: 10.1109/TNSM.2019.2946467.
  15. Sicari, S., Hailes, S., Turgut, D., Sharafeddine, S., Desai, U.B.: Security, privacy and trust management in the internet of things. Ad Hoc Networks 11(8), 2013, pp. 2623– 2624. DOI: http://dx.doi.org/10.1016/j.adhoc.2013.06.006.
  16. Bai, K., Liu, P.: A data damage tracking quarantine and recovery (DTQR) scheme for mission-critical database systems. In: 12th International Conference on Extending Database Technology: Advances in Database Technology, Association for Computing Machinery, U.S.A., 2009, pp. 720–731. DOI:https://doi.org/10.1145/1516360.1516443.
  17. Panda, B., Yalamanchili, R.: Transaction Fusion in the Wake of Information Warfare. In: Proceedings of the 2001 ACM Symposium on Applied Computing, Association for Computing Machinery, Las Vegas, Nevada, USA, 2001, pp. 242—247. DOI:https://doi.org/10.1145/372202.372333.
  18. Ammann, P., Jajodia, S., Liu, P.: Recovery from malicious transactions, IEEE Transactions on Knowledge and Data Engineering 14(5), 2002, pp. 1167-1185. DOI:10.1109/TKDE.2002.1033782.
  19. Haraty, R.A., Saba, R.: Information reconciliation through an agent-controlled graph model. Soft Computing 24(18), 2020, pp. 14019–14037. DOI:https://doi.org/10.1007/s00500-020-04779-x.
  20. Haraty, R.A., Zbib, M., Masud, M.: Data damage assessment and recovery algorithm from malicious attacks in healthcare data sharing systems. Peer to Peer Networking Applications 9(5), 2016, pp. 812–823. DOI: http://dx.doi.org/10.1007/s12083-015-0361-z.
  21. Kaddoura, S., Haraty, R.A., Zekri, A., Masud, M.: Tracking and Repairing Damaged Healthcare Databases Using the Matrix. International Journal Distributed Sensors Networks 11(6), 2016, pp. 1–8. DOI: https://doi.org/10.1155/2015/914305.
  22. Haraty, R.A., El Sai, M.: Information warfare: a lightweight matrix-based approach for database recovery. Knowledge and Information Systems 50(1), 2017, pp. 287–313. DOI: https://doi.org/10.1007/s10115-016-0940-1.
  23. Haraty, R.A., Kaddoura, S., Zekri, A.: Transaction Dependency Based Approach for Database Damage Assessment Using a Matrix. International Journal of Semantic Web and Information Systems 13(2), 2017, pp. 74–86. DOI:10.4018/IJSWIS.2017040105.
  24. Haraty, R.A., Kaddoura, S., Zekri, A.: Recovery of business intelligence systems: Towards guaranteed continuity of patient centric healthcare systems through a matrix-based recovery approach. Telematics and Informatics 35(4), 2018, pp. 801–814. https://doi.org/10.1016/j.tele.2017.12.010.
  25. Kaddoura, S.: Information matrix: Fighting back using a matrix. In: 12th International Conference of Computer Systems and Applications (AICCSA), IEEE, Marrakech, Morocco, 2015, pp. 1-6. DOI:10.1109/AICCSA.2015.7507127.
  26. Boukhari, B.: An Effective Hash Based Assessment and Recovery Algorithm for Healthcare Systems. Masters Thesis. Lebanese American University, Lebanon, 2020.
  27. Chakraborty, A., Majumdar, A., Sural, S.: A column dependency-based approach for static and dynamic recovery of databases from malicious transactions. International Journal of Information Security, 9, 2010, pp. 51–67. DOI:https://doi.org/10.1007/s10207-009-0095-0.
  28. Rao, U.P., Patel, D.R.: Incorporation of Application Specific Information for Recovery in Database from Malicious Transactions. Information Security Journal: A Global Perspective 22(1), 2013, pp. 35–45. DOI: https://doi.org/10.1080/19393555.2013.781721.
  29. Kurra, K., Panda, B., Li, W.-N., Hu, Y.: An Agent Based Approach to Perform Damage Assessment and Recovery Efficiently after a Cyber Attack to Ensure EGovernment Database Security. In: 48th Hawaii International Conference on System Sciences, IEEE, Hawaii, 2015, pp. 2272-–2279. DOI:10.1109/HICSS.2015.272.
  30. El Zarif, O., Haraty, R. A.: Toward information preservation in healthcare systems. Innovation in Health Informatics, Academic Press, 2020, pp. 163–185. DOI:https://doi.org/10.1016/B978-0-12-819043-2.00007-1.

  31.  Introduction. The introduction is poor. The function of the Introduction is to: Establish the context of the work being reported. This is accomplished by discussing the relevant primary research literature (with citations) and summarizing our current understanding of the problem you are investigating;  (Relevant works are not being cited in this Introduction, which gives the impression that they are being ignored.)
    - State the purpose of the work in the form of the hypothesis, question, or problem you investigated; and, 
    - Briefly explain your rationale and approach and, whenever possible, the possible outcomes your study can reveal. 
    - Therefore, I suggest that the Introduction be rewritten, so that the manuscript sounds more interesting for the scientific community. Please rewrite a more complete Introduction following the above described, it is important to thoroughly review the literature and state of the art (please review recent papers, 2018-2021), and cite the most relevant related work.
    - Introduction. Typically, in information security, Cryptographic Hash Functions are used to detect alterations (data corruption) in the information, I consider that in this manuscript it is necessary to cite some relevant works (recent 2018-2021) in this regard and it would be interesting if the authors compare results of the performance of the proposed algorithm versus the well-known Hash Functions. 

The introduction was rewritten to include more citations of relevant work, introduction about information security and cryptographic hash functions. In addition to, an explanation of the problem, approach, and results obtained. The introduction is copied below.

Introduction:

Most of the applications of information systems, especially healthcare systems, are now based on online databases that contain huge data. The security of such databases is essential to ensure an information system that follows the CIA security model: confidentiality, integrity, and availability [1]. Information security involves all processes for protecting data and reducing the adverse effects of any incidents of unlawful use, disclosure, deletion, corruption, or any form of misuse. As defined by [2], information security is the assurance that the information risks and controls are in balance.  Cryptographic hash functions are widely used in information security, in many areas like digital signatures and authentication, cybersecurity for risk management, and healthcare systems security [3-6].

Information warfare can be defined from different perspectives. In this paper, it is defined as an attack on a database to cause a denial of service of the information system [7], while defensive information warfare is the set of actions that protects the database from attacks.

The defensive information warfare paradigm consists of three main layers: prevention, detection, and recovery [8]. Preventive measures are the first line of defense. In this layer, techniques should be designed to guard hardware, software, and user data against threats from both outsiders as well as from malicious insiders. Although many security preventive measures are encountered in all healthcare sectors, still statistics prove that there is always a successful attack [8]. Hence, it is crucial to have detection measures to detect any tampering with data. Detection is usually done by the intrusion detection system (IDS). Once an IDS detects an attack, recovery should start to rollback any changes to ensure the integrity of data. As the world moves towards the IoT era, companies and organizations are replacing traditional paper-based data with new electronic data to achieve higher efficiency in processing. Online data provides availability and ease of sharing, but on the other hand, it poses privacy and security concerns.

This paper mainly focuses on healthcare systems, where data needs to be highly available to doctors and nurses to help them do their job, as downtime may be life-threatening to patients. In addition to the immediate availability of data, it should also be secure and correct. Any intentional or unintentional misinformation can jeopardize patients’ health and potentially lives. Information about patients such as medication, allergies, surgeries, medical conditions, and medical history are confidential and should only be available to authorized personnel. Hence, the CIA triad becomes essential to any implemented system. Last year, more than 8 billion malicious data attacks occurred [9]. In addition, 89% of healthcare organizations experienced data breaches in the past two years [9]. Healthcare databases are a precious target for attackers because they facilitate identity theft and cybercrimes. Any denial of service of healthcare databases affects patients’ lives.

The most critical metric to be considered, while securing healthcare databases, is speed. Patient history should be retrieved as soon as possible, and healthcare transactions should not be affected for a long time by an attack. Hence, database assessment for the healthcare system recovery should be efficient to prevent long downtime while performing these procedures.

The objective of the paper is to develop an efficient algorithm to detect malicious transactions on patient’s data. Given the fact that a malicious transaction may not be immediately detected, and hence, may affect other transactions, dependency among transactions is taken into consideration by the proposed algorithm. The problem statement is further explained in section 2. A parallel algorithm using multithreading is proposed, based on the transaction dependency paradigm. The time complexity of the serial version of the algorithm is O(M+NQ+N^3). Where M is the total number of transactions, N is the number of insertions in transactions to be tested, Q is the time for dependency check. The time complexity is improved to O(Q+N^2) is the parallel version of the algorithm. The memory complexity is O(N+KL), where K is the number of transactions in one area handled by one thread, and L is the number of threads. The algorithm analysis shows an improvement over similar algorithms, namely the HashTable approach [22], and the Single Matrix approach [26]. In terms of time, the proposed algorithm runs up to 4 times faster than compared results, with the worst run up to 3 times faster. In terms of memory, the algorithms save 60,000 bytes and 120,000 bytes over HashTable and Single Matrix approaches, respectively.

The remainder of the paper is structured as follows: Section 2 presents the problem statement. Section 3 gives an overview of related works. In section 4, the proposed algorithm is described along with discussion of its complexity analysis. Section 5 presents and discusses the experimental results of the algorithm. Finally, section 6 presents the conclusion and proposed future enhancements.

  1. State of the art. The authors did not correctly review the literature and are ignoring some relevant papers that can be easily found in this Journal and this Editorial. Which is important to cite and briefly describe the state-of-the-art papers and mention the disadvantages, trends and open problems.

The following references have been added and cited in the introduction and background:

Anderson, J. M.; Why we need a new definition of information security. Computers & Security. 22 (4): 2003, pp. 308–313. doi:10.1016/S0167-4048(03)00407-3.

Ahmed, A.A.; Ahmed, W.A. An Effective Multifactor Authentication Mechanism Based on Combiners of Hash Function over Internet of Things. Sensors 2019, 19, 3663. https://doi.org/10.3390/s19173663

Kure, H.I.; Islam, S.; Razzaque, M.A. An Integrated Cyber Security Risk Management Approach for a Cyber-Physical System. Appl. Sci. 2018, 8, 898. https://doi.org/10.3390/app8060898

Ondiege, B.; Clarke, M.; Mapp, G. Exploring a New Security Framework for Remote Patient Monitoring Devices. Computers 2017, 6, 11. https://doi.org/10.3390/computers6010011

Papamartzivanos, D.; Menesidou, S.A.; Gouvas, P.; Giannetsos, T. A Perfect Match: Converging and Automating Privacy and Security Impact Assessment On-the-Fly. Future Internet 2021, 13, 30. https://doi.org/10.3390/fi13020030

A section (A - Healthcare Information Security and Hash Functions) has been added to the Background section.

A.     Healthcare Information Security and Hash Functions

Cryptographic hash functions are widely used in information security, in many areas like digital signatures and authentication, as an illustration of using hash functions in security, a multi-factor authentication mechanism based on hashing is presented in [3]. Moreover, in [4], an integrated cybersecurity framework for risk management is presented and tested using a power grid system. In healthcare, a framework for remote patient monitoring allowing multiple users from one device is presented in [5]. Another automated method for assessing the impact of privacy and security is presented in [6], based on interdependency graph models and data processing flows.

  1. Section 4 THE MODE, I suggest it be renamed "Proposed Method”, because Model, normally in scientific research represents a Mathematical Model, and in this manuscript, it is not the case.

Section 4 has been renamed to Proposed Method.

  1. Section 4. The method must be described in more detail because it is not clear how the algorithm detects the "contagion" or "virus" in the database? 

Added “Assumptions” section under “Proposed Method”. One of the assumptions is that an intrusion detection system exists and provides the algorithm with the list of malicious transactions.

  1. So it is suggested that the authors clarify in the algorithm what happens when the virus is detected? how long does the quarantine last? How is the correction or healing performed? etc.

Once the intrusion detection system supplies the malicious transactions to our algorithm, the affected transactions are discovered and stored in the testing list until all affected transactions are found. That is when recovery is initiated.

  1. In addition, to clarify the method, it is suggested to add a detailed flow chart that describes the operation of the proposed algorithm.

A detailed flowchart is added to the paper.

  1.  The authors mention that the algorithm works in a similar way to the dynamics of Covid-19, however, in the case of this algorithm, the state of quarantine or isolation is not clear.

Once the intrusion detection system announces that there is a malicious attack, all the system will stop for damage assessment. Thus, all the transactions whether malicious or benign will stop. The damage assessment algorithm will identify the affected transactions and thus will be stopped and retracted. In this way, no other benign transaction can read from these transactions until they are cleaned. This is the concept of isolation.

  1. Experimental Setup. Please specify the version of the Java language used.

Java version is 1.8, specifically 1.8.0_45. This was added under “Experimental Setup” section. Additional information was also added about the libraries used, maven version, and mysql version.

All algorithms tested below are implemented using Java programming language, specifically version 1.8.0_45. For connection to the database, the mysql.mysql-connector-java library (version 8.0.21) is used. For measuring memory consumption of specific objects, the com.carrotsearch.java-sizeof library (version 0.0.5) is used. The code was packaged along with all dependencies using Apache Maven version 3.6.1 into a jar file, which was then run on the different algorithms and with different malicious transaction IDs.

MySQL (version 15.1 distribution 10.4.13-MariaDB)

  1. The quality of all the figures must be improved, they look blurry, it seems like screenshot.

All images were replaced with new ones with higher quality. The ratio was also adjusted for figures that were stretched.

  1. Line 437, the authors say…. The proposed algorithm is faster than the other two algorithms [please put the references]

The other two algorithms in the comparison are mentioned in the beginning of the “Experimental Results” section.

Added reference to [22] and [26], which are the below papers:

Haraty, R.A., El Sai, M.: Information warfare: a lightweight matrix-based approach for database recovery. Knowledge and Information Systems 50(1), 2017, pp. 287–313. DOI: https://doi.org/10.1007/s10115-016-0940-1.

Boukhari, B.: An Effective Hash Based Assessment and Recovery Algorithm for Healthcare Systems. Master’s Thesis. Lebanese American University, Lebanon, 2020.

  1. To verify the dataset and the performance of the proposed algorithm, and with the purpose that other readers and authors can verify and compare the performance of new algorithms, it is suggested that the authors share their dataset as supplementary data, according to the Open Access policies of this publisher.

The dataset used is the Northwind database which is available publicly. The SQL file used to load the data can be provided.

Additional information about the dataset is added in the “Experimental Results” section:

The Northwind database is a sample database used for demonstration purposes of some of Microsoft’s products, especially when it comes to SQL Server. It has been used for experiments in the field of damage assessment and database recovery. The data in the database belongs to a made-up company, named Northwind Traders, and includes information about sales.

A particular version of the Northwind database, written for MySQL syntax, is used. The SQL file is loaded before every test. From the database, only transactions on the below tables are studied in the experiments:

– Employees: employees of the company.

– Suppliers: suppliers of products.

– Customers: customers of the company.

– Categories: categories of products.

– Products: available products.

– Orders: orders placed.

– OrderDetails: details about the order.

The studied relationships between these tables, and constituting the different “areas” are as follows:

- Categories and Products

- Suppliers and Products

- Customers and Orders

- Products and OrderDetails

- Orders and OrderDetails

  1. In addition, the data set should be described in more detail in the manuscript.

Information about the database used, including table and relationships, is added in the “Experimental Results” section. The database used is the Northwind database, a widely known database used in demonstration of features in Microsoft products and in research.

  1. The results presented in Figures 1-7, it is suggested that the authors analyze, discuss, and interpret in greater detail.

We believe that the discussion is detailed. If there is anything missing, or the reviewers can point us to what is missing that would be great. We will gladly add what is needed, but we feel the discussion is rigorous.

  1. It is suggested to add a comparative table of characteristics (features) of the proposed algorithm versus the related work, please analyze and interpret the information shown in the table.

The comparisons between the different algorithms really depend on the data structure that is used, We feel we covered those details in the literature section part.

  1. The conclusions should be updated according to the new results and should emphasize the main results.

There are no new results added and we restudied the conclusion and found it that is stresses the main results.  Again, any explicit pointers will be appreciated.

  1. References. It is recommended to homogenize the reference fields. For better visibility, please add the DOI (when apply) to all References. Other references are missing important fields, as Vol. or Issue, etc. It is important to present the references clearly and with all the required fields. 

References have been revised and missing information completed. References are copied below.

REFERENCES

  1. Stapleton, J.J.: Security without Obscurity: A Guide to Confidentiality, Authentication, and Integrity (1st ed.). Auerbach Publications, 2014, pp. 355. DOI:https://doi.org/10.1201/b16885.
  2. Anderson, J. M.; Why we need a new definition of information security. Computers & Security. 22 (4): 2003, pp. 308–313. doi:1016/S0167-4048(03)00407-3.
  3. Ahmed, A.A.; Ahmed, W.A. An Effective Multifactor Authentication Mechanism Based on Combiners of Hash Function over Internet of Things. Sensors 2019, 19, 3663. https://doi.org/10.3390/s19173663.
  4. Kure, H.I.; Islam, S.; Razzaque, M.A. An Integrated Cyber Security Risk Management Approach for a Cyber-Physical System. Sci. 2018, 8, 898. https://doi.org/10.3390/app8060898
  5. Ondiege, B.; Clarke, M.; Mapp, G. Exploring a New Security Framework for Remote Patient Monitoring Devices. Computers 2017, 6, 11. https://doi.org/10.3390/computers6010011
  6. Papamartzivanos, D.; Menesidou, S.A.; Gouvas, P.; Giannetsos, T. A Perfect Match: Converging and Automating Privacy and Security Impact Assessment On-the-Fly. Future Internet 2021, 13, 30. https://doi.org/10.3390/fi13020030
  7. Hutchinson, B., Hutchinson, W., Warren, M.: Information Warfare: Corporate Attack and Defence in a Digital World. 1st edn. Butterworth-Heinemann, 2001.
  8. Gao, Y., Y. Peng, Feng Xie, Wei Zhao, D. Wang, Xuefeng Han, Tianbo Lu and Z. Li: Analysis of Security Threats and Vulnerability for Cyber-Physical Systems. Proceedings of 2013 3rd International Conference on Computer Science and Network Technology, 2013, pp. 50-55. DOI:https://doi.org/10.1109/ICCSNT.2013.6967062.
  9. Smith, T. T.: Examining Data Privacy Breaches in Healthcare. Walden Dissertations and Doctoral Studies, 2016, 2623. https://scholarworks.waldenu.edu/dissertations/2623.
  10. Ragothaman P., Panda B.: Analyzing Transaction Logs for Effective Damage Assessment. In: Gudes E., Shenoi S. (eds) Research Directions in Data and Applications Security. IFIP — The International Federation for Information Processing, vol 128. Springer, Boston, MA, 2003. DOI:https://doi.org/10.1007/978-0-387-35697-6_8.
  11. Panda, B., Haque, K.A.: Extended Data Dependency Approach: A Robust Way of Rebuilding Database. In Proceedings of the 2002 ACM Symposium on Applied Computing (SAC '02). Association for Computing Machinery, New York, NY, USA, 2002, pp. 446–452. DOI:https://doi.org/10.1145/508791.508875
  12. Dbouk, T., Mourad, A., Otrok, H., Tout, H. and Talhi, C.: A Novel Ad-Hoc Mobile Edge Cloud Offering Security Services Through Intelligent Resource-Aware Offloading. IEEE Transactions on Network and Service Management 16(4), 2019, pp. 1665–1680. DOI:10.1109/TNSM.2019.2939221.
  13. Dbouk, T., Mourad, A., Otrok, H., Talhi, C.: Towards ad-hoc cloud-based approach for mobile intrusion detection. In: 12th International Conference on Wireless and Mobile Computing, Networking and Communications (WiMob), IEEE, Los Alamitos, CA, USA, 2016, pp. 1–8. DOI:http://dx.doi.org/10.1109/WiMOB.2016.7763251.
  14. Kherraf, N., Sharafeddine, S., Assi, C.M. and Ghrayeb, A.: Latency and Reliability Aware Workload Assignment in IoT Networks with Mobile Edge Clouds. IEEE Transactions on Network and Service Management 16(4), 2019, pp. 1435–1449. DOI: 10.1109/TNSM.2019.2946467.
  15. Sicari, S., Hailes, S., Turgut, D., Sharafeddine, S., Desai, U.B.: Security, privacy and trust management in the internet of things. Ad Hoc Networks 11(8), 2013, pp. 2623– 2624. DOI: http://dx.doi.org/10.1016/j.adhoc.2013.06.006.
  16. Bai, K., Liu, P.: A data damage tracking quarantine and recovery (DTQR) scheme for mission-critical database systems. In: 12th International Conference on Extending Database Technology: Advances in Database Technology, Association for Computing Machinery, U.S.A., 2009, pp. 720–731. DOI:https://doi.org/10.1145/1516360.1516443.
  17. Panda, B., Yalamanchili, R.: Transaction Fusion in the Wake of Information Warfare. In: Proceedings of the 2001 ACM Symposium on Applied Computing, Association for Computing Machinery, Las Vegas, Nevada, USA, 2001, pp. 242—247. DOI:https://doi.org/10.1145/372202.372333.
  18. Ammann, P., Jajodia, S., Liu, P.: Recovery from malicious transactions, IEEE Transactions on Knowledge and Data Engineering 14(5), 2002, pp. 1167-1185. DOI:10.1109/TKDE.2002.1033782.
  19. Haraty, R.A., Saba, R.: Information reconciliation through an agent-controlled graph model. Soft Computing 24(18), 2020, pp. 14019–14037. DOI:https://doi.org/10.1007/s00500-020-04779-x.
  20. Haraty, R.A., Zbib, M., Masud, M.: Data damage assessment and recovery algorithm from malicious attacks in healthcare data sharing systems. Peer to Peer Networking Applications 9(5), 2016, pp. 812–823. DOI: http://dx.doi.org/10.1007/s12083-015-0361-z.
  21. Kaddoura, S., Haraty, R.A., Zekri, A., Masud, M.: Tracking and Repairing Damaged Healthcare Databases Using the Matrix. International Journal Distributed Sensors Networks 11(6), 2016, pp. 1–8. DOI: https://doi.org/10.1155/2015/914305.
  22. Haraty, R.A., El Sai, M.: Information warfare: a lightweight matrix-based approach for database recovery. Knowledge and Information Systems 50(1), 2017, pp. 287–313. DOI: https://doi.org/10.1007/s10115-016-0940-1.
  23. Haraty, R.A., Kaddoura, S., Zekri, A.: Transaction Dependency Based Approach for Database Damage Assessment Using a Matrix. International Journal of Semantic Web and Information Systems 13(2), 2017, pp. 74–86. DOI:10.4018/IJSWIS.2017040105.
  24. Haraty, R.A., Kaddoura, S., Zekri, A.: Recovery of business intelligence systems: Towards guaranteed continuity of patient centric healthcare systems through a matrix-based recovery approach. Telematics and Informatics 35(4), 2018, pp. 801–814. https://doi.org/10.1016/j.tele.2017.12.010.
  25. Kaddoura, S.: Information matrix: Fighting back using a matrix. In: 12th International Conference of Computer Systems and Applications (AICCSA), IEEE, Marrakech, Morocco, 2015, pp. 1-6. DOI:10.1109/AICCSA.2015.7507127.
  26. Boukhari, B.: An Effective Hash Based Assessment and Recovery Algorithm for Healthcare Systems. Master’s Thesis. Lebanese American University, Lebanon, 2020.
  27. Chakraborty, A., Majumdar, A., Sural, S.: A column dependency-based approach for static and dynamic recovery of databases from malicious transactions. International Journal of Information Security, 9, 2010, pp. 51–67. DOI:https://doi.org/10.1007/s10207-009-0095-0.
  28. Rao, U.P., Patel, D.R.: Incorporation of Application Specific Information for Recovery in Database from Malicious Transactions. Information Security Journal: A Global Perspective 22(1), 2013, pp. 35–45. DOI: https://doi.org/10.1080/19393555.2013.781721.
  29. Kurra, K., Panda, B., Li, W.-N., Hu, Y.: An Agent Based Approach to Perform Damage Assessment and Recovery Efficiently after a Cyber Attack to Ensure EGovernment Database Security. In: 48th Hawaii International Conference on System Sciences, IEEE, Hawaii, 2015, pp. 2272-–2279. DOI:10.1109/HICSS.2015.272.
  30. El Zarif, O., Haraty, R. A.: Toward information preservation in healthcare systems. Innovation in Health Informatics, Academic Press, 2020, pp. 163–185. DOI:https://doi.org/10.1016/B978-0-12-819043-2.00007-1.

  31. After reviewing the manuscript, I consider that the contribution to the state of the art is still not clear, therefore, I suggest that the authors add another metric to assess its performance of the proposed algorithm, for example, they can use Halstead Complexity Analysis (Halstead's metrics) and compare the results versus the related work or another equivalent metric to evaluate algorithms.

We have followed what previous works and other authors have done in the past to validate their work, and we followed the same methodology, even the same public database to compare the findings. We have compared the algorithms accordingly, (damage assessment time and recovery slots consumption). We are not sure that adding more metrics would validate the results further.

I hope these comments and suggestions help to improve the quality of the manuscript and clarify the contribution to the state of the art. In case that the authors decide to resubmit the manuscript, please highlight all changes in another color in the revised version of the manuscript.

Thank you wholeheartedly for reading the paper very closely and making the constructive comments. The effort is really appreciated.

Reviewer 2 Report

The paper "A Parallelized Database Damage Assessment Approach after Cyberattacks for Healthcare Systems" offers a novel multithreading-based algorithm for damage assessments in healthcare systems composed of interlinked databases. Although this computational development's contribution is evident concerning time consumption by faster detecting  potential affected transactions due to parallelism, the limitation resides on memory efficiency, in which case the algorithm performs poorly due to larger Java threads memory consumption. Nevertheless, the proposal performs sufficiently well when the environment is controlled to single-threaded algorithms. The English writing needs minor proofreading before an eventual publication, not to be drastically changed. There are conceptual and discussion errors in the manuscript, and the citation and reference need cohesion. The paper meets the scope of the journal. My comments are numbered as follows:

  1. I cannot understand the citation-reference structure the authors have adopted. They are not numbered according to first appearance, or ordered alphabetically. Many references are missing citations (e.g. references [1], [12], [13], among others, are missing citations). Please check all of this, citing and referencing the manuscript properly.

  1. Consider expanding the literature review in section "3. Background" to include papers about cybersecurity in healthcare, which is the context of this application. MDPI journals have a lot material on this.

  1. I would like to see a brief discussion on how the algorithm would perform when concurrent operations of multi-transactions are interleaved. In addition, real world applications involve some independent (non-linked) or small databases, and also the detection efficiency of the algorithm would depend on how many times users access potential data/transactions. I can understand simulating those environments may be out of the current scope, but some discussion according to the authors' perspective can add value to the assessment.

  1. What do you mean about "rigorously serializable" (line 229)? Schedules can be serial or non-serial. I believe you mean "no transaction Ti ∀ i = 1, 2, 3, …, J starts until a running transaction has ended". If this is the case, please reformulate this fourth assumption.

  1. Including a visual framework (e.g. a flow diagram) to summarize the reasoning in the last paragraph of "4.2. Proposed Algorithm" or, especially, for the "4.4. Example" would considerably improve your paper. It can be exhaustive keep returning to the transactions definitions to infer the example.

  1. How is the initial list of malicious (or the first one) detected and provided to start the algorithm? How can you guarantee the initial infected transaction is not recursive to other potential infected transactions?

  1. Define "thread" in the pseudo-code before designating it to terminate. Also, i and i (in italic) are the same variables? i (italic) is out of the loop flow to perform the repetitions.

  1. The sub-section "4.4 Example" is incorrect. T5 writes E blindly. T3 is potentially infected, so it should update the testing list array (lines 297-298).

  1. Please provide information on how the testing list array containing the affected transactions is robust to duplicates. The authors discuss in the third paragraph of 4.5 an alternative, but it is not clear if they implemented it.

  1. Discuss the managerial implications on the claimed memory efficiency and queuing thread optimization. It is not clear how these prospects can be inferred in a daily healthcare operation.

  1. Definitions need a reference. Have a read and cite definitions when adequate.

  1. Implementing the 3 algorithms 10 times on each malicious transaction may fail to produce statistical-supported metrics for generalizing any interlinked database results. Why this amount? Is there any time-consuming constraint or other constraint considered by the authors but not reported in the simulation setup?

  1. Please summarize the results for the time, affected transactions, data and memory efficiency in an informative table, comparing descriptive statistics (mean, median, quartiles, range, standard deviation…) and algorithms' prospects.

  1. Figures report the results for Haraty et al. (2020), not Boukhari (2019) discussed in the text. Please provide the correct information, discussion, or implementation.

  1. In conclusion, I would suggest trying to link with healthcare systems in the current pandemics.

After these revisions, I will be glad to recommend this publication.

Author Response

Reviewer 2:

The paper "A Parallelized Database Damage Assessment Approach after Cyberattacks for Healthcare Systems" offers a novel multithreading-based algorithm for damage assessments in healthcare systems composed of interlinked databases. Although this computational development's contribution is evident concerning time consumption by faster detecting potential affected transactions due to parallelism, the limitation resides on memory efficiency, in which case the algorithm performs poorly due to larger Java threads memory consumption. Nevertheless, the proposal performs sufficiently well when the environment is controlled to single-threaded algorithms. The English writing needs minor proofreading before an eventual publication, not to be drastically changed. There are conceptual and discussion errors in the manuscript, and the citation and reference need cohesion. The paper meets the scope of the journal. My comments are numbered as follows:

  1. I cannot understand the citation-reference structure the authors have adopted. They are not numbered according to first appearance or ordered alphabetically. Many references are missing citations (e.g., references [1], [12], [13], among others, are missing citations). Please check all of this, citing and referencing the manuscript properly.

The references have been updated. References that are nor cited (1, 9, 12, 13, 27 in the old manuscript) have been removed, and numbering of all references has been updated to be in ascending order.

REFERENCES

  1. Stapleton, J.J.: Security without Obscurity: A Guide to Confidentiality, Authentication, and Integrity (1st ed.). Auerbach Publications, 2014, pp. 355. DOI:https://doi.org/10.1201/b16885.
  2. Anderson, J. M.; Why we need a new definition of information security. Computers & Security. 22 (4): 2003, pp. 308–313. doi:1016/S0167-4048(03)00407-3.
  3. Ahmed, A.A.; Ahmed, W.A. An Effective Multifactor Authentication Mechanism Based on Combiners of Hash Function over Internet of Things. Sensors 2019, 19, 3663. https://doi.org/10.3390/s19173663.
  4. Kure, H.I.; Islam, S.; Razzaque, M.A. An Integrated Cyber Security Risk Management Approach for a Cyber-Physical System. Sci. 2018, 8, 898. https://doi.org/10.3390/app8060898
  5. Ondiege, B.; Clarke, M.; Mapp, G. Exploring a New Security Framework for Remote Patient Monitoring Devices. Computers 2017, 6, 11. https://doi.org/10.3390/computers6010011
  6. Papamartzivanos, D.; Menesidou, S.A.; Gouvas, P.; Giannetsos, T. A Perfect Match: Converging and Automating Privacy and Security Impact Assessment On-the-Fly. Future Internet 2021, 13, 30. https://doi.org/10.3390/fi13020030
  7. Hutchinson, B., Hutchinson, W., Warren, M.: Information Warfare: Corporate Attack and Defence in a Digital World. 1st edn. Butterworth-Heinemann, 2001.
  8. Gao, Y., Y. Peng, Feng Xie, Wei Zhao, D. Wang, Xuefeng Han, Tianbo Lu and Z. Li: Analysis of Security Threats and Vulnerability for Cyber-Physical Systems. Proceedings of 2013 3rd International Conference on Computer Science and Network Technology, 2013, pp. 50-55. DOI:https://doi.org/10.1109/ICCSNT.2013.6967062.
  9. Smith, T. T.: Examining Data Privacy Breaches in Healthcare. Walden Dissertations and Doctoral Studies, 2016, 2623. https://scholarworks.waldenu.edu/dissertations/2623.
  10. Ragothaman P., Panda B.: Analyzing Transaction Logs for Effective Damage Assessment. In: Gudes E., Shenoi S. (eds) Research Directions in Data and Applications Security. IFIP — The International Federation for Information Processing, vol 128. Springer, Boston, MA, 2003. DOI:https://doi.org/10.1007/978-0-387-35697-6_8.
  11. Panda, B., Haque, K.A.: Extended Data Dependency Approach: A Robust Way of Rebuilding Database. In Proceedings of the 2002 ACM Symposium on Applied Computing (SAC '02). Association for Computing Machinery, New York, NY, USA, 2002, pp. 446–452. DOI:https://doi.org/10.1145/508791.508875
  12. Dbouk, T., Mourad, A., Otrok, H., Tout, H. and Talhi, C.: A Novel Ad-Hoc Mobile Edge Cloud Offering Security Services Through Intelligent Resource-Aware Offloading. IEEE Transactions on Network and Service Management 16(4), 2019, pp. 1665–1680. DOI:10.1109/TNSM.2019.2939221.
  13. Dbouk, T., Mourad, A., Otrok, H., Talhi, C.: Towards ad-hoc cloud-based approach for mobile intrusion detection. In: 12th International Conference on Wireless and Mobile Computing, Networking and Communications (WiMob), IEEE, Los Alamitos, CA, USA, 2016, pp. 1–8. DOI:http://dx.doi.org/10.1109/WiMOB.2016.7763251.
  14. Kherraf, N., Sharafeddine, S., Assi, C.M. and Ghrayeb, A.: Latency and Reliability Aware Workload Assignment in IoT Networks with Mobile Edge Clouds. IEEE Transactions on Network and Service Management 16(4), 2019, pp. 1435–1449. DOI: 10.1109/TNSM.2019.2946467.
  15. Sicari, S., Hailes, S., Turgut, D., Sharafeddine, S., Desai, U.B.: Security, privacy and trust management in the internet of things. Ad Hoc Networks 11(8), 2013, pp. 2623– 2624. DOI: http://dx.doi.org/10.1016/j.adhoc.2013.06.006.
  16. Bai, K., Liu, P.: A data damage tracking quarantine and recovery (DTQR) scheme for mission-critical database systems. In: 12th International Conference on Extending Database Technology: Advances in Database Technology, Association for Computing Machinery, U.S.A., 2009, pp. 720–731. DOI:https://doi.org/10.1145/1516360.1516443.
  17. Panda, B., Yalamanchili, R.: Transaction Fusion in the Wake of Information Warfare. In: Proceedings of the 2001 ACM Symposium on Applied Computing, Association for Computing Machinery, Las Vegas, Nevada, USA, 2001, pp. 242—247. DOI:https://doi.org/10.1145/372202.372333.
  18. Ammann, P., Jajodia, S., Liu, P.: Recovery from malicious transactions, IEEE Transactions on Knowledge and Data Engineering 14(5), 2002, pp. 1167-1185. DOI:10.1109/TKDE.2002.1033782.
  19. Haraty, R.A., Saba, R.: Information reconciliation through an agent-controlled graph model. Soft Computing 24(18), 2020, pp. 14019–14037. DOI:https://doi.org/10.1007/s00500-020-04779-x.
  20. Haraty, R.A., Zbib, M., Masud, M.: Data damage assessment and recovery algorithm from malicious attacks in healthcare data sharing systems. Peer to Peer Networking Applications 9(5), 2016, pp. 812–823. DOI: http://dx.doi.org/10.1007/s12083-015-0361-z.
  21. Kaddoura, S., Haraty, R.A., Zekri, A., Masud, M.: Tracking and Repairing Damaged Healthcare Databases Using the Matrix. International Journal Distributed Sensors Networks 11(6), 2016, pp. 1–8. DOI: https://doi.org/10.1155/2015/914305.
  22. Haraty, R.A., El Sai, M.: Information warfare: a lightweight matrix-based approach for database recovery. Knowledge and Information Systems 50(1), 2017, pp. 287–313. DOI: https://doi.org/10.1007/s10115-016-0940-1.
  23. Haraty, R.A., Kaddoura, S., Zekri, A.: Transaction Dependency Based Approach for Database Damage Assessment Using a Matrix. International Journal of Semantic Web and Information Systems 13(2), 2017, pp. 74–86. DOI:10.4018/IJSWIS.2017040105.
  24. Haraty, R.A., Kaddoura, S., Zekri, A.: Recovery of business intelligence systems: Towards guaranteed continuity of patient centric healthcare systems through a matrix-based recovery approach. Telematics and Informatics 35(4), 2018, pp. 801–814. https://doi.org/10.1016/j.tele.2017.12.010.
  25. Kaddoura, S.: Information matrix: Fighting back using a matrix. In: 12th International Conference of Computer Systems and Applications (AICCSA), IEEE, Marrakech, Morocco, 2015, pp. 1-6. DOI:10.1109/AICCSA.2015.7507127.
  26. Boukhari, B.: An Effective Hash Based Assessment and Recovery Algorithm for Healthcare Systems. Master’s Thesis. Lebanese American University, Lebanon, 2020.
  27. Chakraborty, A., Majumdar, A., Sural, S.: A column dependency-based approach for static and dynamic recovery of databases from malicious transactions. International Journal of Information Security, 9, 2010, pp. 51–67. DOI:https://doi.org/10.1007/s10207-009-0095-0.
  28. Rao, U.P., Patel, D.R.: Incorporation of Application Specific Information for Recovery in Database from Malicious Transactions. Information Security Journal: A Global Perspective 22(1), 2013, pp. 35–45. DOI: https://doi.org/10.1080/19393555.2013.781721.
  29. Kurra, K., Panda, B., Li, W.-N., Hu, Y.: An Agent Based Approach to Perform Damage Assessment and Recovery Efficiently after a Cyber Attack to Ensure EGovernment Database Security. In: 48th Hawaii International Conference on System Sciences, IEEE, Hawaii, 2015, pp. 2272-–2279. DOI:10.1109/HICSS.2015.272.
  30. El Zarif, O., Haraty, R. A.: Toward information preservation in healthcare systems. Innovation in Health Informatics, Academic Press, 2020, pp. 163–185. DOI:https://doi.org/10.1016/B978-0-12-819043-2.00007-1.

  1. Consider expanding the literature review in section "3. Background" to include papers about cybersecurity in healthcare, which is the context of this application. MDPI journals have a lot of material on this.

The following references have been added and cited in the introduction and background:

Anderson, J. M.; Why we need a new definition of information security. Computers & Security. 22 (4): 2003, pp. 308–313. doi:10.1016/S0167-4048(03)00407-3.

Ahmed, A.A.; Ahmed, W.A. An Effective Multifactor Authentication Mechanism Based on Combiners of Hash Function over Internet of Things. Sensors 2019, 19, 3663. https://doi.org/10.3390/s19173663

Kure, H.I.; Islam, S.; Razzaque, M.A. An Integrated Cyber Security Risk Management Approach for a Cyber-Physical System. Appl. Sci. 2018, 8, 898. https://doi.org/10.3390/app8060898

Ondiege, B.; Clarke, M.; Mapp, G. Exploring a New Security Framework for Remote Patient Monitoring Devices. Computers 2017, 6, 11. https://doi.org/10.3390/computers6010011

Papamartzivanos, D.; Menesidou, S.A.; Gouvas, P.; Giannetsos, T. A Perfect Match: Converging and Automating Privacy and Security Impact Assessment On-the-Fly. Future Internet 2021, 13, 30. https://doi.org/10.3390/fi13020030

A section (A - Healthcare Information Security and Hash Functions) has been added to the Background section.

A.     Healthcare Information Security and Hash Functions

Cryptographic hash functions are widely used in information security, in many areas like digital signatures and authentication, as an illustration of using hash functions in security, a multi-factor authentication mechanism based on hashing is presented in [3]. Moreover, in [4], an integrated cybersecurity framework for risk management is presented and tested using a power grid system. In healthcare, a framework for remote patient monitoring allowing multiple users from one device is presented in [5]. Another automated method for assessing the impact of privacy and security is presented in [6], based on interdependency graph models and data processing flows.

  1. I would like to see a brief discussion on how the algorithm would perform when concurrent operations of multi-transactions are interleaved. In addition, real world applications involve some independent (non-linked) or small databases, and the detection efficiency of the algorithm would depend on how many times users access potential data/transactions. I can understand simulating those environments may be out of the current scope, but some discussion according to the authors' perspective can add value to the assessment.

The experimental setup and execution for this algorithm is very similar to previous works published on information recovery. We just followed similar suit in order to have a fair assessment and comparison of the result.

  1. What do you mean about "rigorously serializable" (line 229)? Schedules can be serial or non-serial. I believe you mean "no transaction Ti ∀ i = 1, 2, 3, …, J starts until a running transaction has ended". If this is the case, please reformulate this fourth assumption.

A fourth assumption, as indicated in the comment, is added as to the list of assumptions.

  1. Including a visual framework (e.g., a flow diagram) to summarize the reasoning in the last paragraph of "4.2. Proposed Algorithm" or, especially, for the "4.4. Example" would considerably improve your paper. It can be exhaustive keep returning to the transactions definitions to infer the example.

The flowchart below was added to Section 4. Where the algorithm starts by checking transactions in the testing list and iterates until no more transactions remain. If the current transaction in the testing list, other transactions in the area that interact with the current transactions are identified and added to the testing list. Otherwise, waiting on testing list takes place until new elements are encountered. This process continues until no more transactions need to be tested.

  1. How is the initial list of malicious (or the first one) detected and provided to start the algorithm? How can you guarantee the initial infected transaction is not recursive to other potential infected transactions?

Added “Assumptions” section under “Proposed Method”. One of the assumptions is that an intrusion detection system exists and provides the algorithm with the list of malicious transactions. Other assumptions include transactions having unique sequential IDs according to chronological order and the DBMS scheduler history being rigorously serializable (no transaction Ti reads from transaction Tj if j>i).

  1. Define "thread" in the pseudo-code before designating it to terminate. Also, i and i(in italic) are the same variables? (italic) is out of the loop flow to perform the repetitions.

Updated pseudocode to define threads, fixed formatting of variables:

???????_???? ← list of malicious transaction(s)

threads <- list of threads covering the various areas

Each thread will run the below steps:

? ← 0

while not all threads are waiting on ???????_???? do

  if i < size(testing_list) then

    ???????_??????????? ←???????_???? [?]

    if ???????_??????????? ∈ area of thread then

      find transactions in area that interacted with     current_transaction

      append these transactions to testing_list if not already in it

    else

      skip current transaction since out of thread’s area

    endif

    ? ← ? + 1

  else

    wait on testing_list for new elements

  endif

endwhile

  1. The sub-section "4.4 Example" is incorrect. T5 writes E blindly. T3 is potentially infected, so it should update the testing list array (lines 297-298).

T3 is affected by the malicious transaction T1 since T1 wrote C and T3 reads from C. However, T5 blindly writes E, and E is not written by any of the previous transactions following and including the malicious transaction T1. Hence, E is not infected and transaction T5 is unaffected by the attack. That is why it does not show in the testing_list. 

  1. Please provide information on how the testing list array containing the affected transactions is robust to duplicates. The authors discuss in the third paragraph of 4.5 an alternative, but it is not clear if they implemented it.

This is discussed in the “Complexity Analysis” section, quoted text below:

“It is ensured that the list does not have duplicates, so the algorithm loops over the list, before the insertion, to check whether the transaction exists or not. This makes the complexity of insertion into the list O(n) where n is the size of the list. Being a simple way to ensure no duplicate transactions in the testing list, this approach is used in our implementation. Another solution to prevent duplicates is working on the thread area to remove a transaction ID once this transaction has been processed for the first time. This will prevent processing the same transaction twice. Thus, the final list can then be cleared of duplicates, if any.”

The approach used in the current implementation is simply to loop over the testing_list for every insertion and ensure we are not inserting a duplicate entry. 

  1. Discuss the managerial implications on the claimed memory efficiency and queuing thread optimization. It is not clear how these prospects can be inferred in a daily healthcare operation.

The main discussion about threads and overhead memory is that threads require a certain amount of memory simply to exist and run. The studied database has a relatively small number of transactions, and particularly a small number of transactions per area (or per thread). Therefore, the memory overhead from threads adds to the overall memory consumption of the algorithm, making our algorithm seem less memory-efficient than the other 2 algorithms. The table that shows only the data structure memory (ignoring thread memory) shows that, in fact, the actual memory needed by our algorithm (ignoring thread overhead) is far less. There is also a table that compares the memory for a single-threaded algorithm, also showing that the number of threads is what is causing the overall memory reported to be higher than expected.

The argument is that, once the number of transactions to be studied by the algorithm becomes larger, the memory needed by threads does not increase. If the number of transactions is sufficiently large, the thread memory may become negligible with respect to the memory consumed by needed data structures. Since our algorithm requires less memory for data structures than other approaches (which need quadratic memory), it will perform better on huge databases with a large number of transactions.

Below is part of the “Interpretation” subsection of the paper:

Our algorithm primarily targets large databases that have huge numbers of transactions. With such scenarios, the parallelism gained from threads would be much more evident, making our algorithm much faster than other approaches. In addition, when databases are large with many tables and transactions, it is less likely to be caught in the worst case of having many affected transactions grouped into very few areas. As for memory, the memory consumed by threads tends to be negligible when compared with the memory consumed by algorithm data structures when huge number of transactions are involved.

With lots of transactions, our algorithm needs linear memory because the extra thread memory can be assumed constant. Hence, it will need much less memory than other approaches that require quadratic memory, which would be catastrophic with millions of transactions for example. 

  1. Definitions need a reference. Have a read and cite definitions when adequate.

We went and reread the paper and fixed the references and added the correct ones when needed.

  1. Implementing the 3 algorithms 10 times on each malicious transaction may fail to produce statistical-supported metrics for generalizing any interlinked database results. Why this amount? Is there any time-consuming constraint or other constraint considered by the authors but not reported in the simulation setup?

Updated the “Experimental Setup” to better explain why we chose 10 runs per algorithm.

 Since the time taken by the algorithm may differ, due to the load on the system, the response time of the database, cores available, etc., all the algorithms run 10 times for each malicious transaction ID. Among those runs, the best, worst, and average time were recorded. The memory required is generally not affected by other factors and hence has one reporting which was the same for all runs.

The number 10 was chosen to ensure that other factors mentioned above, such as other processes on the system, availability or cores, or database response time, do not greatly affect the reported results or provide a disadvantage to some algorithm. We understand that when measuring small units such as microseconds, the results may vary between runs, so we assume the average time reported to be the most accurate comparison. Going beyond 10 runs does not considerably change the reported results.

  1. Please summarize the results for the time, affected transactions, data, and memory efficiency in an informative table, comparing descriptive statistics (mean, median, quartiles, range, standard deviation…) and algorithms' prospects.

We have summarized our findings like other previous works that have been published in the literature, and we even followed the same format of comparison (i.e., damage assessment time, and memory slots consumption). We do not believe adding extra descriptive statistics would add anything substantial to the findings of the experimental results.

  1. Figures report the results for Haraty et al. (2020), not Boukhari (2019) discussed in the text. Please provide the correct information, discussion, or implementation.

Figures changed to have “Boukhari (2020)” instead of “Haraty et al. (2020)”. 

  1. In conclusion, I would suggest trying to link with healthcare systems in the current pandemics.

We could not agree with you more. We added the following to the conclusion: “Healthcare databases are a prized target for attackers because they enable identity theft and cybercrimes. Any denial of service of healthcare databases affects patients’ lives.

The gravest metric to be considered, while securing healthcare databases, is pace. Patient history should be recovered as soon as possible, and healthcare transactions should not be affected for a long time by an attack. Hence, database assessment for the healthcare system recovery should be effective to prevent extended stoppage while acting these measures.”

After these revisions, I will be glad to recommend this publication.

Round 2

Reviewer 1 Report

I have carefully reviewed the revised version of the manuscript and I was able to verify that the authors have responded satisfactorily most of my comments and suggestions, the quality of the manuscript has been improved, the contribution to the state of the art is clear.

Only, I have a minor concern, I consider that the authors should correctly cite the Reference of Northwind database used for this paper.

I thanks the authors who have responded satisfactorily to these comments and suggestions.

Author Response

We have added a reference to the Northwind database:

31. Northwind Database. (2011). In Codeplex. Retrieved March 26, 2021, from https://northwinddatabase.codeplex.com/.